METHODS AND RESOURCES

# SCRINSHOT enables spatial mapping of cell states in tissue sections with single-cell resolution

Alexandros Sountoulidis[1,2]*, Andreas Liontos[1,2], Hong Phuong Nguyen[1,2], Alexandra B. Firsova[1,2], Athanasios Fysikopoulos[3], Xiaoyan Qian[1,4], Werner Seeger[3], Erik Sundström[5], Mats Nilsson[1,4], Christos Samakovlis[1,2,3]*

**1** Science for Life Laboratory, Solna, Sweden, **2** Department of Molecular Biosciences, Wenner-Gren Institute, Stockholm University, Stockholm, Sweden, **3** Molecular Pneumology, Cardiopulmonary Institute, Justus Liebig University, Giessen, Germany, **4** Department of Biochemistry and Biophysics, Stockholm University, Stockholm, Sweden, **5** Department of Neurobiology, Care Sciences and Society, Karolinska Institutet, Stockholm, Sweden

* christos.samakovlis@scilifelab.se (CS); alexandros.sountoulidis@scilifelab.se (AS)

**Data Availability Statement:** The datasets and analysis files of the current study have been deposited to Zenodo repository (DOI: 10.5281/zenodo.3634561 and 10.5281/zenodo.3978632).

## Abstract

Changes in cell identities and positions underlie tissue development and disease progression. Although single-cell mRNA sequencing (scRNA-Seq) methods rapidly generate extensive lists of cell states, spatially resolved single-cell mapping presents a challenging task. We developed SCRINSHOT (Single-Cell Resolution IN Situ Hybridization On Tissues), a sensitive, multiplex RNA mapping approach. Direct hybridization of padlock probes on mRNA is followed by circularization with SplintR ligase and rolling circle amplification (RCA) of the hybridized padlock probes. Sequential detection of RCA-products using fluorophore-labeled oligonucleotides profiles thousands of cells in tissue sections. We evaluated SCRINSHOT specificity and sensitivity on murine and human organs. SCRINSHOT quantification of marker gene expression shows high correlation with published scRNA-Seq data over a broad range of gene expression levels. We demonstrate the utility of SCRINSHOT by mapping the locations of abundant and rare cell types along the murine airways. The amenability, multiplexity, and quantitative qualities of SCRINSHOT facilitate single-cell mRNA profiling of cell-state alterations in tissues under a variety of native and experimental conditions.

## Introduction

Recent advances in single-cell RNA sequencing (scRNA-Seq) technologies enabled transcriptome analysis of individual cells and the identification of new cellular states in healthy and diseased conditions [1]. These methods, however, fail to capture the spatial cellular organization in tissues due to cell dissociation. New spatial transcriptomic methods aim to circumvent the problem of lost cellular topology [2]. They can be divided into 2 categories: first, targeted methods that directly detect specific mRNAs with single-cell resolution such as in situ sequencing (ISS) [3], multiplexed error-robust fluorescence in situ hybridization (MERFISH) [4], and

All scripts are available at: https://github.com/AlexSount/SCRINSHOT Single cell RNA Sequencing data of AT2 cells (Liu et al., 2019) are freely accessible from authors at Gene Expression Omnibus (GEO) repository (GSE118891). The annotated AT2 samples (according to the provided metadata information) were used in the present study. Single cell RNA Sequencing data of tracheal cells (Montoro et al., 2018) are freely accessible from authors at Gene Expression Omnibus (GEO) repository (GSE103354).

**Funding:** The study was founded by the Swedish Research Council (Vetenskapsrådet, 2016-05059) and the Swedish Cancer Society (Cancerfonden, 160499) to CS. This project has received funding from the European Union's Horizon 2020 research and innovation programme under the grant agreement No. 874656. The funders had no role in study design, data collection and analysis, decision to publish, or preparation of the manuscript.

**Competing interests:** I have read the journal's policy and the authors of this manuscript have the following competing interests: MN and XQ hold shares in CARTANA AB, a company that commercializes in situ sequencing technology.

**Abbreviations:** AI, artificial intelligence; AT2, alveolar type 2; AmpFISH, high-fidelity amplified FISH; CC10, club cell marker Scgb1a1; cDNA, complementary DNA; CGRP, calcitonin gene-related peptide; clampFISH, click-amplifying FISH; FACS, fluorescence-activated cell sorting; FISH, fluorescence in situ hybridization; ISS, in situ sequencing; MERFISH, multiplexed error-robust fluorescence in situ hybridization; NA, not-annotated; NE, neuroendocrine; NEB, neuroepithelial body; osmFISH, ouroboros smFISH method; P1, postnatal day 1; PBVC-1, *Paramecium bursaria* chlorella virus 1; PFA, paraformaldehyde-fixed; PLISH, proximity ligation in situ hybridization; RCA, rolling circle amplification; RFP, red fluorescent protein; ROI, region of interest; RT, reverse transcription; scRNA-Seq, single-cell RNA sequencing; seqFISH, sequential fluorescence in situ hybridization; SG, submucosal gland; smFISH, single-molecule fluorescence in situ hybridization; SNAIL, specific amplification of nucleic acids via intramolecular ligation; STARmap, spatially-resolved transcript amplicon readout mappingTm, melting temperature; UNG, uracil-N-glycosylase; φ29, *Bacillus subtilis* phage phi29 DNA polymerase.

ouroboros smFISH method (osmFISH) [5], and second, global methods, which are based on barcode-annotated positions and next-generation sequencing to resolve RNA topology. The spatial resolution of global methods is still larger than the typical cellular dimensions [6, 7].

Targeted methodologies are based on nucleic acid probes (mainly DNA), complementary to the RNA species of interest, as in all in situ hybridization assays [8]. Single-molecule fluorescence in situ hybridization (smFISH) is the most powerful among the spatial transcriptomic methods and has been used to supplement scRNA-Seq data with spatial information. It utilizes multiple fluorophore-labeled probes, which recognize the same RNA molecule along its length and visualize single RNA molecules as bright fluorescent dots [9, 10]. Nonetheless, this method still retains some limitations such as low signal-to-noise ratio, reduced sensitivity on short transcripts, false positive signals due to unspecific binding of the labeled probes, and low capacity for multiplex detection of many RNA molecules [2, 11, 12]. Multiplex detection with smFISH was initially addressed by the sequential fluorescence in situ hybridization (seqFISH) [13, 14] and the MERFISH [4]. These approaches utilize sequential rounds of hybridization of FISH probes or barcode-based primary probes to detect multiple RNA species. The outstanding throughput of these methods makes them strong candidates for generation of spatial transcriptome maps in tissues. However, because the principle of these techniques is similar to smFISH, they require large number of gene-specific probes, confocal or super-resolution microscopy to deconvolve the signals, and complicated algorithms for both probe design and analysis. Nevertheless, the low signal-to-noise ratio still remains a major technical challenge of these methods, especially for tissue sections with strong autofluorescence from structural extracellular matrix components like collagen and elastin [15]. New strategies for signal amplifications, such as branched-DNA amplification (RNAScope) [16] and hybridization chain reaction [17, 18], have been recently combined with sophisticated probe design (high-fidelity amplified FISH [AmpFISH]) [19] to increase sensitivity and specificity of smFISH.

Padlock probes have been successfully used to detect RNA species [20]. These are linear DNA molecules, with complementary arms to the target mRNA sequence and a common "backbone." Upon hybridization with the target sequence, they can be ligated, creating circular single-stranded DNA molecules, which are used as templates for signal amplification using *Bacillus subtilis* phage phi29 DNA polymerase (Φ29) polymerase-mediated rolling circle amplification (RCA) [21]. RCA products are long single-stranded DNA molecules containing hundreds of copies [22] of the complementary padlock probe sequence. They can be detected with fluorophore-labeled-oligos, which recognize either their RNA-specific sequence or their backbone. Because each RCA product contains hundreds of repeats of the same detected sequence, the signal-to-noise ratio increases significantly, facilitating signal detection by conventional epifluorescence microscopy. Also, multiplexity has been integrated into the method by sequencing by ligation [23, 24]. Because commercial ligases such as T4 DNA ligase, T4 RNA ligase 2, and Ampligase show low activity on DNA/RNA hybrids, RNA has to be reverse-transcribed to complementary DNA (cDNA) fragments before introducing padlock probes [3]. However, cDNA synthesis on fixed tissue sections is a challenging and expensive procedure [11], which prompted new, elegant approaches to circumvent reverse transcription (RT). For example, click-amplifying FISH (clampFISH) utilizes click-chemistry to ligate DNA probes after hybridization on their RNA targets [25]. More recently, "H-type" DNA probes, which are hybridized to both RNA and padlock probes (proximity ligation in situ hybridization [PLISH] [12]), or specific amplification of nucleic acids via intramolecular ligation (SNAIL)-design probes have been successfully used to facilitate intramolecular ligation of the padlock probes (spatially resolved transcript amplicon readout mapping [STARmap]) [24]).

*Paramecium bursaria* chlorella virus 1 (PBVC-1) DNA ligase (also known as SplintR ligase) shows strong ligase activity of DNA sequences in DNA/RNA hybrids. This enzyme is a DNA

ligase encoded by PBVC1, first discovered by Ho and colleagues [26]. A number of studies have successfully applied SplintR ligase for RNA species detection in cultured cells [11, 27–29]. However, the fidelity of SplintR ligation in end-joining is questionable because it tolerates mismatches at the padlock probe junction, and its usefulness is debated. In addition, SplintR ligase shows 1% of its ligation activity on a 1-nucleotide gap of a nicked duplex DNA substrate [11, 30].

We present SCRINSHOT, an optimized protocol for multiplex RNA in situ detection on paraformaldehyde-fixed (PFA) tissue sections. We validated the sensitivity, specificity, and multiplexity of SCRINSHOT and showed that it is quantitative over a broad range of gene expression levels. It is based on the in situ sequencing protocol [3] but bypasses the costly and inefficient RT on fixed tissue to gain higher detection efficiency, by utilizing SplintR ligase. To minimize false positive artifacts, we utilized 40-nucleotide-long target-specific sequences in the padlock probes in combination with stringent hybridization conditions. SCRINSHOT performs on a variety of tissues including lung, kidney, and heart and readily detects several epithelial, endothelial, and mesenchymal cells. We tested the multiplexing of SCRINSHOT on mouse and human tissue sections, by simultaneous detection of characterized cell-type-selective markers. SCRINSHOT successfully identified distinct cell types and helped to create spatial maps of large tissue areas at single-cell resolution.

## Results and discussion

### SCRINSHOT overview

SCRINSHOT evolved from our attempts to improve the detection sensitivity and reduce the cost of the in situ sequencing method, using PFA-fixed material. PFA fixation significantly improves the histology but makes RNA less accessible to enzymes and padlock probes [11, 23]. We focused on stringent padlock probe hybridization to RNA targets and omitted the inefficient in situ cDNA synthesis step (Fig 1 and S1 Fig). Gene-specific padlock probes are first directly hybridized to 40nt long, unique sequences on the mRNA targets. Upon ligation of bound padlock probes, we amplify the padlock probe sequence using Φ29 polymerase dependent RCA. RCA products are subsequently detected by fluorophore-labeled oligos, which recognize the gene-specific part of the padlock, as previously described [3]. Multiplexity is reduced compared with sequencing by ligation [23], which theoretically allows the detection of 256 transcript species in 4 hybridization runs, or to other barcode-based approaches [4, 11]. To increase the number of detected genes, we used sequential hybridization cycles of fluorophore-labeled, uracil-containing oligos. After each detection cycle, fluorescent probes are removed by enzymatic fragmentation by uracil-N-glycosylase (UNG) and stringent washes. Sequential detection also enables the separate detection of low, medium, and high abundant RNA species, increasing the dynamic range of detection [31]. Our detection probes were each labeled by 1 of 3 commonly used fluorophores and allowed up to 10 hybridization and imaging cycles, typically detecting 30 genes. After image acquisition, we utilized manual segmentation of nuclei and open access analysis tools for image stitching and signal quantification to construct a simple pipeline for quantitative mapping of expression counts for 20–30 genes to thousands of cells on tissue sections. The detailed protocol starting with tissue fixation and leading to mapping is presented in the S1 Text.

### SCRINSHOT specificity depends on stringent hybridization of the padlock probe

The specificity of SCRINSHOT crucially relies on the precise targeting of the SplintR ligation activity to the correct sites of the interrogated mRNA. A recent study [30] reported that the

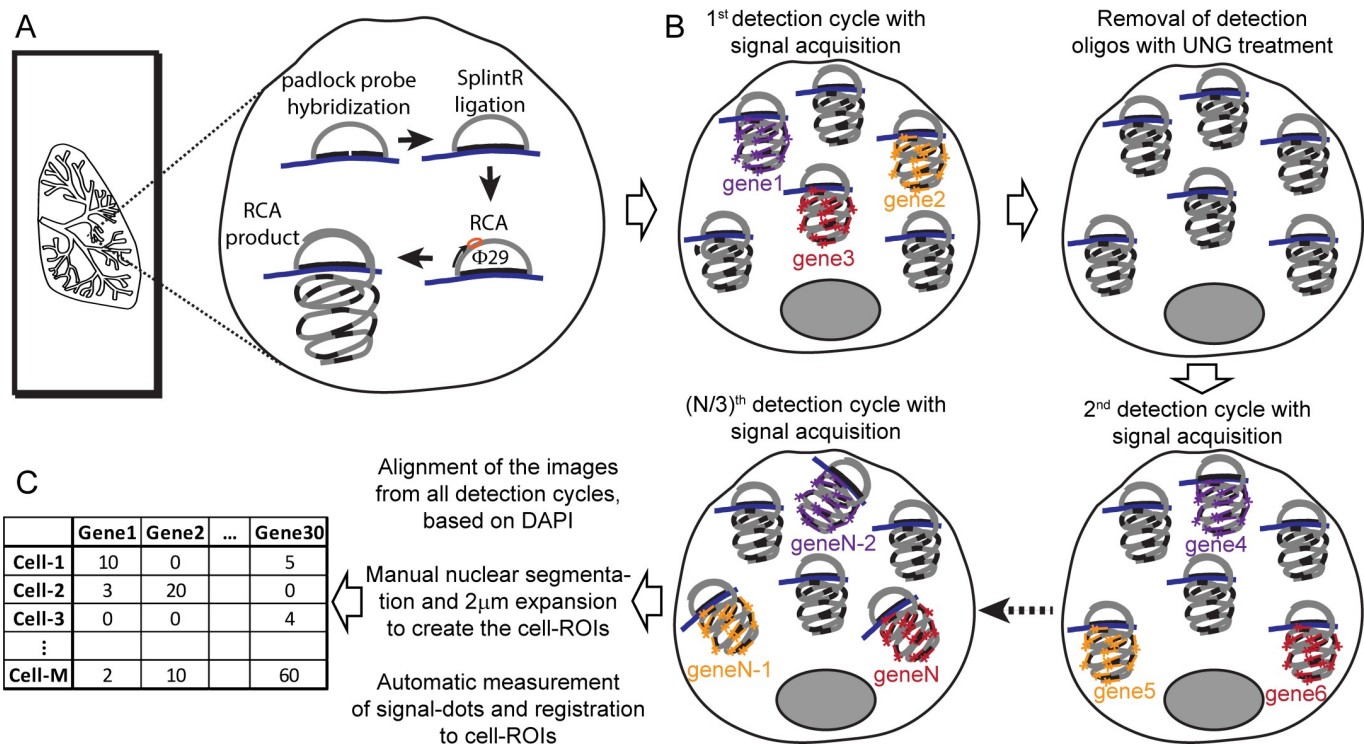

**Fig 1. Schematic representation of SCRINSHOT.** The major steps of the assay are (A) all padlock probe hybridization for all the targeted RNA species followed by ligation and RCA-amplification. (B) RCA-products are detected sequentially reading 3 per cycle, with FITC-, Cy3-, and Cy5-labeled detection oligos, which recognize the gene-specific part of the padlock probes. (C) Images from all detection cycles are aligned using DAPI nuclear staining and segmented to create the cell ROIs, and signal dots are counted and registered to the cell ROIs. RCA, rolling circle amplification; ROI, region of interest; UNG, uracil-N-glycosylase; φ29, *Bacillus subtilis* phage phi29 DNA polymerase.

fidelity of this ligase is poor as it tolerates mismatches at padlock probe junctions. In addition, SplintR shows 1% of its ligation activity on a 1-nucleotide gap of a nicked duplex DNA substrate, and it is unable to join ends across a 2-nucleotide gap [32]. These results question the specificity of SplintR ligase-based methods for in situ RNA detection. We reasoned that the choice of padlocks with high melting points (melting temperature [Tm] around 70°C) followed by stringent washes after DNA/RNA hybridization would circumvent SplintR promiscuity. We tested the dependence of SCRINSHOT signals on the hybridization and ligation steps of the padlock by generating mutant padlocks, predicted to affect either the hybridization of the 3′-padlock arm or the sequence of the ligation site. The 3′-scrambled arm of *Scgb1a1* padlock is expected to fail in hybridizing with the *Scgb1a1* mRNA, resulting in a linear, unligated padlock and therefore in a block in RCA. The single-mismatch probe contains a single replaced nucleotide at the ligation site in 5′-end (C > G) of *Scgb1a1* padlock. This substitution was designed to address the effect of promiscuous ligation on the signal. In the same experiment, we included a slide in which we omitted the SplintR ligase from the reaction mixture to test whether padlock hybridization alone is sufficient to generate some RCA. In all tested conditions, we also used an *Actb* normal padlock probe. This transcript was detected at similar levels in all slides and served as a control for the reactions with the mutated padlocks. We first counted the dots of *Scgb1a1* and *Actb* signals in all airway cells of sequential lung tissue sections and plotted their ratios at the different conditions (Fig 2). The signal from single-mismatch padlock probe was reduced by 20%, compared with the normal *Scgb1a1* probe demonstrating the low SplintR fidelity for the sequence of the ligation site (Fig 2B, 2B′ and 2E).

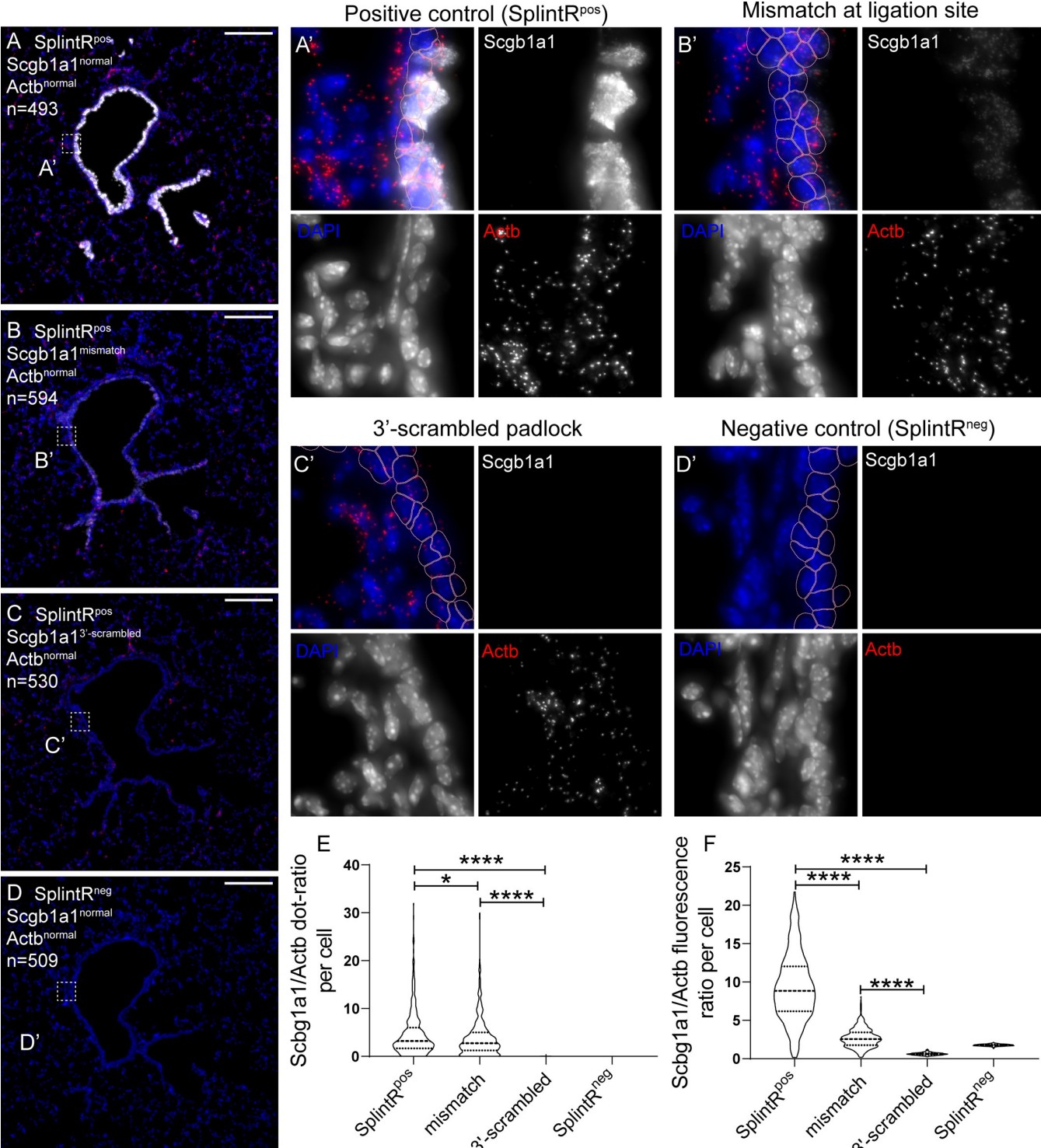

**Fig 2. SCRINSHOT specificity relies on stringent hybridization of padlock probes to their target RNAs.** Images of SCRINSHOT signal, using normal *Scgb1a1* padlock probe (A), a *Scgb1a1* padlock probe with a point mutation at its ligation site (B), a *Scgb1a1* padlock probe with 3′-scrambled arm (C) and normal padlock probe but omitting SplintR ligase (D). *Actb* normal padlock probe was used in all conditions as internal control. DAPI: blue, *Scgb1a1*: gray, *Actb*: red. "n" indicates the number of airway cells in the corresponding images. (A'-D') Magnified areas of the indicated positions (square brackets) of images in the left. Pink outlines show the 2-μm expanded airway nuclear ROIs, which are considered as cells. Scale bar: 150 μm. (E) Violin plot of the *Scgb1a1* and *Actb* signal dots ratio in all airway cells. The ratio of cells with zero *Actb*-dots was considered as zero. There were no statistically significant differences for *Actb* signal dots, between the analyzed conditions, except for the SplinR$^{neg}$, that gave no signal, as expected, indicating that ligation is obligatory for signal detection (*P*

values for *Actb* dot comparisons: "SplintR[pos] versus mismatch" = 0.7706, "SplintR[pos] versus scrambled" = 0.8837, "SplintR[pos] versus SplintR[neg]" ≤ 0.0001, "mismatch versus scrambled" = 0.6523, "mismatch versus SplintR[neg]" ≤ 0.0001, "scrambled versus SplintR[neg]" ≤ 0.0001). (F) Violin plot of the *Scgb1a1* and *Actb* fluorescence intensity ratio in all airway cells. SplintR[pos] $n = 473$, mismatch $n = 574$, 3′-scrambled $n = 507$ and SplintR[neg] $n = 488$. The data underlying this Figure can be found in 10.5281/zenodo.3634561. ROI, region of interest.

Similarly, we introduced point mutations in the padlock probe ligation sites of *Cyp2f2* [33], another club cell–selective gene and of *Etv5*, an alveolar type 2 epithelial cell (AT2) marker [34]. The C>T exchange in the 3′-arm of the *Cyp2f2* padlock probe caused 50% reduction of the detected dots (S2A–S2D Fig). All of the *Cyp2f2* signal was localized in airway cells, in contrast to the signal from *Actb*, which was detected without positional preference, as expected. Thus, the *Cyp2f2* signal distribution provides additional evidence that the specificity of the SCRINSHOT is predominantly grounded on the hybridization of the padlock probe arm sequences and not by the 2 nucleotides in the ligation site. Similarly, the introduction of a T>C mutation in the 5′-arm of *Etv5* padlock probe reduced the signal by 10-fold, as indicated by the average number of *Etv5* dots in more than 800 alveolar cells of the same area, in 2 sequential sections. The signal from the *Actb* internal control showed no statistically significant differences (S2E–S2H Fig).

We noticed that the *Scgb1a1* signal showed considerable crowding and potential saturation, leading to underestimation of the total number of RCA-products for this highly expressed gene. This crowding was evident even when we used 5-fold less *Scgb1a1* padlock probe (0.01 μM) compared with all other padlock probes. In an attempt to more accurately quantify the differences between the signals from normal and mutated *Scgb1a1* padlocks, we measured the overall fluorescence intensity (raw integrated density) of the airway cell regions of interest (ROIs). This showed that the single-mismatch at the ligation site of the *Scgb1a1* padlock probe causes 3-fold fluorescence signal reduction (Fig 2F).

The observed signal saturation prompted us to further examine the possible causes of saturation, because it affects the accurate measurement of detected RNA transcripts. First, analysis of different airway positions showed that *Scgb1a1* signal saturation was more evident in club cells of distal airways that express high levels of the marker but not in club cells of proximal airways [35] (S3A Fig). This suggests that saturation depends on the expression levels of the target gene. In addition, the SCRINSHOT signals of *Etv5* and *Cyp2f2* showed no saturation (controls in S2 Fig) correlating with their lower expression levels. In scRNA-Seq experiments, the *Scgb1a1* average $\log_2$(counts+1) was 17.4 in club cells, whereas the *Cyp2f2* average $\log_2$(counts +1) was 14.4, and the *Etv5* average $\log_2$(counts+1) was 9.4 in AT2 cells [36]. These results show that saturation is probe-independent and stems from high expression levels of genes like *Scgb1a1*. We also estimated how SCRINSHOT dot size may influence signal saturation. The average size for an *Etv5* dot was 0.55 $\mu m^2$ and for *Cyp2f2*, 0.57 $\mu m^2$. Because the average cell ROI size was 44.5 $\mu m^2$ (2-μm expanded nucleus), we expect that the maximum number of dots that can evenly fit in the mean cell area without overlap is approximately 80 (S3B Fig). We propose that reduction of the number and concentration of the padlock probes for highly expressed genes can solve or alleviate the problem. The *Scgb1a1* signal was lost when we used the 3′-scrambled padlock, indicating that padlock probe hybridization is necessary for circularization and subsequent RCA (Fig 2C, 2C′ and 2E). The omission of SplintR from the ligation mixture resulted in undetectable signal for both *Scgb1a1* and *Actb* indicating the central role of ligation in signal amplification (Fig 2D, 2D′ and 2E).

In summary, we confirmed the previously reported tolerance of SplintR ligase for single-nucleotide mismatches at the ligation site [26, 30, 37]. Our experimental data show that the specificity of SCRINSHOT largely depends on the stringent hybridization of the 20-nucleotide-long padlock arms (40 nucleotides in total) and less on the specificity of the used ligase.

## A *Scgb1a1* antisense oligonucleotide competes with padlock probe hybridization and signal detection

If the padlock hybridization is the critical step for signal generation, then competition by an oligonucleotide, which recognizes the binding to the mRNA, is expected to proportionally reduce the detected RCA signal. We used the *Scgb1a1* and *Actb* padlocks together with increasing concentrations of a competing, unlabeled oligonucleotide complementary to the *Scgb1a1* mRNA sequence, recognized by the *Scgb1a1* padlock. Inclusion of the competitor reduced the *Scgb1a1* signal in a dose-dependent manner. Equal molar ratios of padlock probe and competitor caused signal reduction by 10-fold, and the signal was eliminated when 5-fold excess of the competitor was used (Fig 3). This suggests that the SCRINSHOT signal is proportional to the

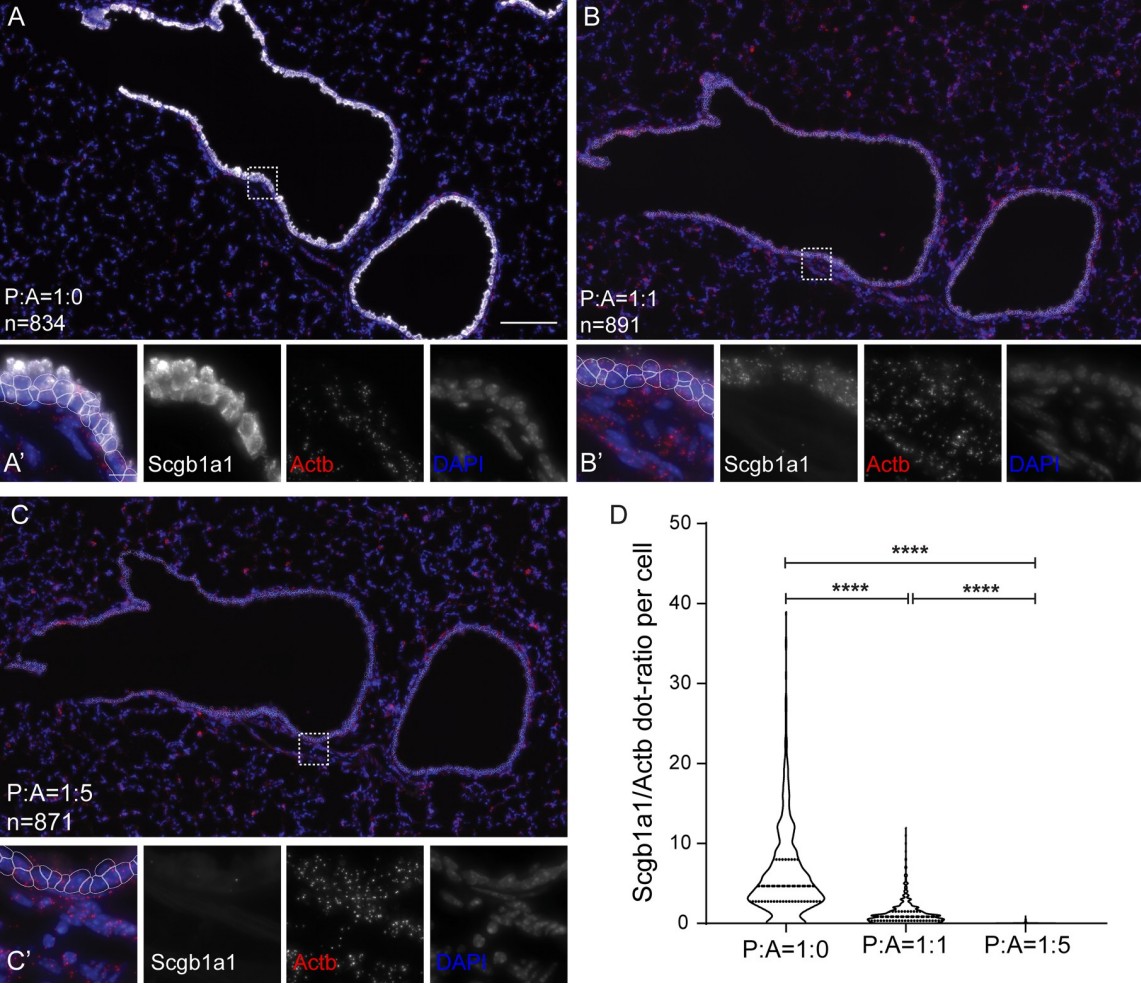

**Fig 3. Concentration-dependent SCRINSHOT detection efficiency.** (A-C) Representative images of SCRINSHOT signals for *Scgb1a1* padlock probes in the absence or presence of an antisense competitor oligo, targeting the binding site of *Scgb1a1* padlock probe. SCRINSHOT signal for *Scgb1a1* padlock probe in the absence of the antisense competitor oligo (A), when mixed in equal concentration with the antisense competitor oligo (B) and when mixed with 5 times higher concentration of the antisense competitor oligo (C). (A'-C') Magnified areas of the indicated positions (square brackets) of images A-C. (D) Violin plot of the *Scgb1a1* and *Actb* signal dots ratio in all airway cells of the 3 compared conditions. Scale bars in (A-C) is 100 μm and (A'-C') is 10 μm. All analyses were done using raw images and the same thresholds. For visualization purposes, brightness and contrast of *Scgb1a1* were set independently in the compared conditions to show the existence of signal in the presence of antisense competitor and avoid signal saturation in its absence. The data underlying this Figure can be found in 10.5281/zenodo.3634561. A, antisense competitor; P, padlock probe.

target expression levels because it can be proportionally competed with increasing concentrations of a synthetic oligo masking the hybridization site. It also highlights the importance of proper padlock design to achieve similarly high Tm values and hybridizations conditions for the different probes.

## Application of SCRINSHOT in other organs

To evaluate SCRINSHOT applicability to other tissues, we performed the assay using PFA-fixed sections from adult mouse kidney and heart and human embryonic lung. In the heart, we examined SCRINSHOT performance for detection of endothelial cell markers, including padlock probes for the endothelial genes *Pecam1*, *Cldn5*, and *Cdh5*, the smooth muscle marker *Acta2*, and the epithelial cell markers *Scgb1a1* and *Sftpc*, as negative controls (S4A Fig). As expected, we detected combinatorial expression of the endothelial markers at the internal surface of the vessels, where endothelial cells are localized. *Acta2* was detected on thick wall vessels, where vascular smooth muscle cells are localized. Also, we detected scattered endothelial cells in the myocardium, as previously described [38] (arrow in S4A Fig). For the kidney, we targeted *Actb* as a generic marker, *Scgb1a1* as a club cell marker, *Sftpc* and *Napsa* as AT2 cell markers, and *Lyz2* as a marker for AT2 cells, macrophages, and neutrophils (S4B Fig) [39]. As expected, *Actb* was uniformly expressed, whereas *Scgb1a1* and *Sftpc* were undetectable. *Lyz2* was expressed by a few scattered cells, which presumably correspond to macrophages [40, 41]. In a subset of kidney tubular structures, we detected *Napsa*, which agrees with the previously described immunohistochemical detection of the marker in renal proximal tubule cells [42].

We also used human embryonic lung sections and probes targeting transcripts encoding 3 transcription factors, *SOX2*, *SOX9*, and *ASCL1* (S5 Fig), which have previously been detected by antibody staining in subsets of lung epithelial cells [43, 44]. In agreement with the published results, the SCRINSHOT signal for *SOX2* was confined mainly in the proximal part of the branching epithelium, whereas *SOX9* was selectively expressed in the distal tips. *ASCL1* expression overlapped with *SOX2*. These experiments show that SCRINSHOT can be readily applied to map cell-type heterogeneity in a variety of tissues.

## SCRINSHOT generates quantitative gene expression profiles in single cells

We further evaluated SCRINSHOT in a *Sftpc-CreER;Rosa-Ai14* reporter mouse expressing red fluorescent protein (RFP), in AT2 cells, upon Tamoxifen induction of the Cre recombinase [45]. In this strain, Cre recombines out a transcriptional/translational STOP cassette [46] of the Rosa26 locus and allows RFP protein expression and fluorescence in AT2 cells [47]. We injected pups with Tamoxifen on postnatal day 1 (P1) and analyzed their lungs on day P21. In the same experiment, we used P21 lungs from *Sftpc-CreER*[neg]-*Rosa-Ai14*[pos] mice, as controls for the RFP induction and lungs from wild-type mice, sacrificed at P60 to assess potential tissue autofluorescence.

Sftpc immunostaining and RFP fluorescence analysis of lung sections from *Sftpc-CreER*[neg]; *Rosa-Ai14*[pos] and *Sftpc-CreER*[pos];*Rosa-Ai14*[pos] mice showed that 3.06% of the 1,111 Sftpc-positive cells were RFP-fluorescence-negative (false negative for RFP fluorescence) and 1.1% of the 1,088 RFP-fluorescence-positive cells were negative for Sftpc staining (false positive for RFP fluorescence) (Fig 4A). RFP false negatives can be attributed to suboptimal efficiency of Cre induction and RFP false positives to leaky activation of Cre and *Rosa* locus (details in https://www.jax.org/strain/007914) or differentiation of AT2 to AT1 cells [48, 49]. This analysis, indicates that the RFP reporter, in this experiment, labels true Sftpc-positive cells with 97% fidelity.

In sections of the same tissue, we first examined the ability of SCRINSHOT to detect the *RFP* mRNA in RFP fluorescent cells. We analyzed 14,167 cells in a large area including airways

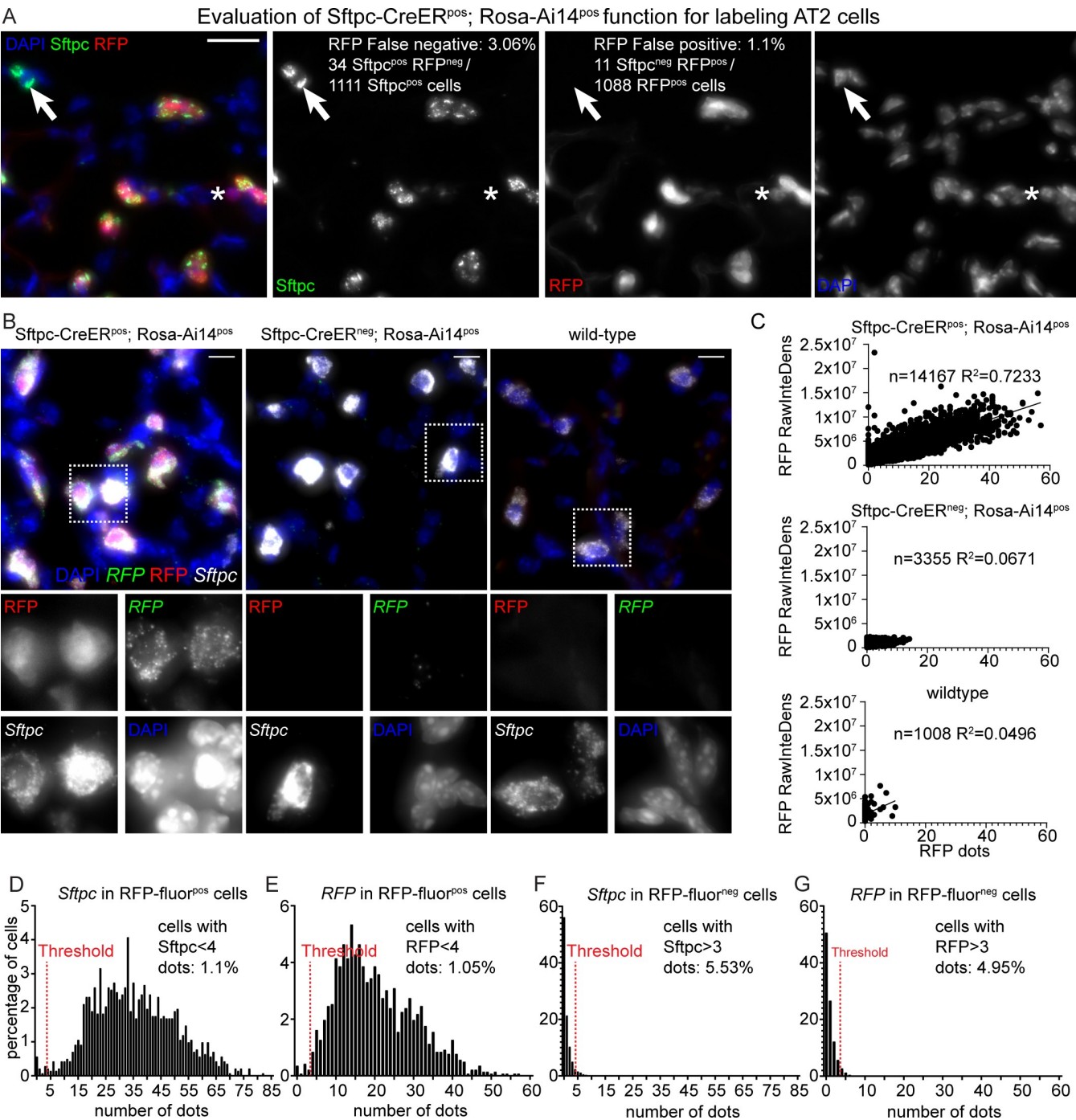

**Fig 4. SCRINSHOT efficiently detects *RFP* and *Sftpc* mRNAs, upon *Sftpc-CreER* driven recombination in *Sftpc-CreER*pos;*Rosa-Ai14*pos lung cells.** (A) Evaluation of recombination efficiency, in the SCRINSHOT-analyzed, P21 *Sftpc-CreER*pos;*Rosa-Ai14*pos lung, after one Tamoxifen induction, on P1. Sftpc protein was detected in AT2 cells with antibody staining and Ai14 (RFP), based on its protein fluorescence. Arrow indicates an Sftpcpos RFPneg AT2 cell and the asterisk, an Sftpcneg RFPpos cell. DAPI: blue, Sftpc-antibody: green, RFP transgene: red. Scale bar: 20 μm. The data underlying this Figure section can be found in 10.5281/zenodo.3978632. (B) Representative images of alveolar regions from P21 *Sftpc-CreER*pos;*Rosa-Ai14*pos, P21 *Sftpc-CreER*neg;*Rosa-Ai14*pos, and P60 wild-type (*Sftpc-CreER*neg;*Rosa-Ai14*neg) lung sections. DAPI: blue, *RFP* mRNA detected with SCRINSHOT: green, RFP protein fluorescence: red and *Sftpc* mRNA detected with SCRINSHOT: white. Square brackets indicate the cropped areas of the bottom part of the panel. Scale bar: 10 μm. (C) Linear regression analyses of RFP protein fluorescence (RAW integrated density) and *RFP* SCRINSHOT dots, in the 3 analyzed lung sections, using GraphPad Prism. The "n" indicates the number of total cells in the analyzed images and "$R^2$" shows the correlation coefficients. (D-G) Histograms of SCRINSHOT results for *Sftpc* and *RFP* in RFP protein fluorescence positive and negative cells. The y-axis shows the percentage of positive cells and x-axis the bins of the detected RNA transcripts number per cell. Threshold was set to 3 dots and shown with red dotted lines. Percentages in "D" and "E" correspond to the cells below threshold

(false negatives) and in "F" and "G" to the cells above threshold (false positives) in RFP protein fluorescence positive and negative cells, respectively. The data underlying these Figure sections can be found in 10.5281/zenodo.3634561 and 10.5281/zenodo.3978632. AT2, alveolar type 2; P21, postnatal day 21; RawInteDens, raw integrated density; RFP, red fluorescent protein.

and alveoli. The RFP fluorescence intensity (raw integrated density) per cell correlated significantly with the number of SCRINSHOT-detected *RFP* mRNA molecules ($R^2 = 0.7233$, Fig 4C). By contrast, there was no correlation between the *RFP* SCRINSHOT signal and the RFP fluorescence ($R^2 = 0.0671$) in 3,355 cells of the *Sftpc-CreER*neg-*Rosa-Ai14*pos control lung. As expected, in the 1,008 analyzed wild-type cells, the correlation of RFP fluorescence and SCRINSHOT dots was very low. Upon closer inspection, most of the signal in this lung was due to high autofluorescence of red blood cells (Fig 4C). We conclude that the *RFP* SCRINSHOT signal directly correlates with the activation of the fluorescent reporter.

To provide a quantitative estimation about the specificity and sensitivity of our method, we used the protein RFP fluorescence as "ground truth" and separated the alveolar cells in RFP fluorescence-positive and -negative cells. Considering that Cre-mediated recombination is under the control of the endogenous *Sftpc* promoter, we measured the detected *Sftpc* and *RFP* mRNA transcripts in both groups. In 1,425 RFP fluorescence positive cells, 1.1% contained 0–3 *Sftpc* dots (Fig 4D), and 1.05% contained 0–3 *RFP* dots (Fig 4E), which were considered as false negative.

Among the 6,848 RFP fluorescence negative cells, 5.53% were positive for *Sftpc* and 4.95% for *RFP* SCRINSHOT dots and were considered as false positives (Fig 4F and 4G). Overall, the inducible knock-in reporter analysis shows that the SCRINSHOT *RFP* and *Sftpc* signals highly correlate with RFP fluorescence intensity, indicating high sensitivity and specificity. The high specificity is additionally supported by the absence of *RFP* SCRINSHOT signal in the analyzed wild-type lung (Fig 4B) of the same experiment.

## Multiplex performance and gene expression quantification using SCRINSHOT

To further explore the utility of SCRINSHOT in the spatial identification of cell types, we first tested whether multiple rounds of hybridization and detection, in lung tissue sections, lead to loss of detection signal for the genes of interest, as seen in other spatial transcriptomics experiments [5]. We selected *Calca*, a known neuroendocrine (NE) gene marker, because of its highly localized expression in neuroepithelial bodies (NEBs) [50]. We detected the *Calca* RCA products during the first and the eighth cycles of hybridization and detection and found no significant loss of signal or decreased specificity. Our analysis indicates that *Calca* SCRINSHOT signal spatially correlates with *Ascl1*, another NE cell marker [51] (S6 Fig). *Ascl1* and *Scgb1a1* SCRINSHOT signals were complementary but also showed some spatial overlap. This overlap likely stems from segmentation faults caused by the high density and interdigitation of NEB cells, as indicated by confocal microscopy analysis of the same tissue area (S7A Fig). Alternatively, low *Scgb1a1* expression could be detected in NE cells as was also published in scRNA-Seq datasets [52] (S8D-S8F Fig). To further investigate this, we performed an independent immunofluorescence analysis of the club cell marker Scgb1a1 (CC10), the NE cell marker calcitonin gene-related peptide (CGRP), and the epithelial cell border marker E-cadherin in NEBs and confirmed that there is Scgb1a1 and Calca signal overlap (arrows in S7B Fig), even with optimal confocal microscopy settings. The highly co-localized SCRINSHOT signals for *Ascl1* and *Calca* and their largely complementary pattern with *Scgb1a1* are in full agreement with the protein expression patterns of the same analyzed markers, further supporting the specificity of the method.

To test SCRINSHOT multiplex performance, we successfully detected, on the same tissue section, 13 additional genes that encode soluble secreted proteins (*Napsa*, *Lgi3*, *Sftpc*, and

*Lyz2*), cell surface proteins and receptors (*Cd74*, *Cldn18*, *Fgfr2*, and *Ager*), the metabolic enzyme *Cyp2f2*, the transgene *RFP* and signaling proteins and transcription factors (*Axin2*, *Spry2*, and *Etv5*) (Fig 5A).

We first used *Sftpc* [53], *Lyz2* [49], *Etv5* [34], and *Napsa* [54], as a panel of AT2-selective markers and performed pairwise correlations of each marker with the rest and with the *RFP* SCRINSHOT dots and RFP fluorescence intensity. We also included the club cell markers *Scgb1a1* and *Cyp2f2* [33], as negative controls. The Spearman correlation coefficients ($\rho$) of the comparisons between *Sftpc*, *Napsa*, *Etv5*, and *RFP* ranged between 0.81 and 0.9, indicating high correlation. *Lyz2* correlates well with the aforementioned AT2 markers, but there are *Lyz2* single positive cells that likely correspond to alveolar macrophages [39]. The highest correlation score ($\rho = 0.93$) was between *Scgb1a1* and *Cyp2f2* because of their exclusive expression by club airway cells. As expected, their expression anticorrelated (negative $\rho$ values) with the AT2 markers and the RFP fluorescence (Fig 5B). This indicates that SCRINSHOT can quantitatively detect multiple selective markers in single cells of a complex tissue.

We also based our evaluation on moderately expressed genes, like *Etv5*, encoding a transcription factor, that are specifically expressed in AT2 cells [34]. We classified the analyzed cells according to their *Etv5* positivity. The analyzed 1,431 cells were scored as *Etv5*$^{pos}$ (>3 dots), whereas the remaining 12,736 were *Etv5*$^{neg}$. *Sftpc* mRNA transcripts were detected in 99% of the *Etv5*$^{pos}$ cells. Similarly, 94.8% of the *Etv5*$^{pos}$ cells were *RFP* positive (Fig 5C). This analysis indicates that SCRINSHOT's specificity is high (95%), even for moderately expressed genes (e.g., *Etv5*), allowing efficient cell-type identification.

In a complementary test, we analyzed a published SmartSeq2 scRNA-Seq dataset of AT2 and club cells (GSE118891) [36]. There, 96.8% of the AT2 cells have more *Etv5* scRNA-Seq counts than the club cells in the same dataset, and 12% of them express low levels of the gene (less than 1 standard deviation from average number of counts). This low-expressing subset of AT2 cells may reflect technical variation or real, biological variability [36]. On the protein level, only 50% of the adult mouse AT2 cells are positive for Etv5 [34]. Thus, if SCRINSHOT were as sensitive as single-cell sequencing by SmartSeq2, we would expect that 3%–12% of the RFP fluorescence positive cells would lack *Etv5* signal dots. Our counting did not detect *Etv5* dots in 3.5% of the 1,425 RFP fluorescence positive cells. Additionally, 14.8% of the RFP fluorescence positive cells had 0–2 *Etv5* dots, which is less than the standard deviation of the average *Etv5* dots (S9 Fig). These results provide additional evidence that SCRINSHOT is in precise agreement with scRNA-Seq methods and that its sensitivity is also high for moderately expressed genes.

To evaluate the utility of SCRINSHOT in complementing scRNA-Seq data with spatial information, we compared the SCRISNSHOT analysis with the aforementioned SmartSeq2 dataset, focusing on the annotated 156 AT2 cells [36]. We used AT2-related genes, excluding *Ascl1*, *Calca*, and *Scgb1a1* and analyzed their SCRINSHOT signals in RFP fluorescence positive cells. The mean values of SCRINSHOT dots in RFP fluorescence positive cells were proportional to the scRNA-Seq raw count values of AT2 cells. Spearman correlation analysis showed a strong correlation between the results of the 2 methodologies ($\rho = 0.951$) for highly, moderately, and lowly expressed genes (Fig 5D). The proportional mean gene expression levels in SCRINSHOT and scRNA-Seq argue that SCRINSHOT provides a suitable alternative for rapid, in situ evaluation of cell states, detected by scRNA-Seq.

## Spatial mapping of tracheal cell heterogeneity using SCRINSHOT

Recently, 2 studies addressed the heterogeneity of tracheal epithelium using scRNA-Seq. These studies identified a new pulmonary cell type, which expresses *Cftr* and therefore considered to

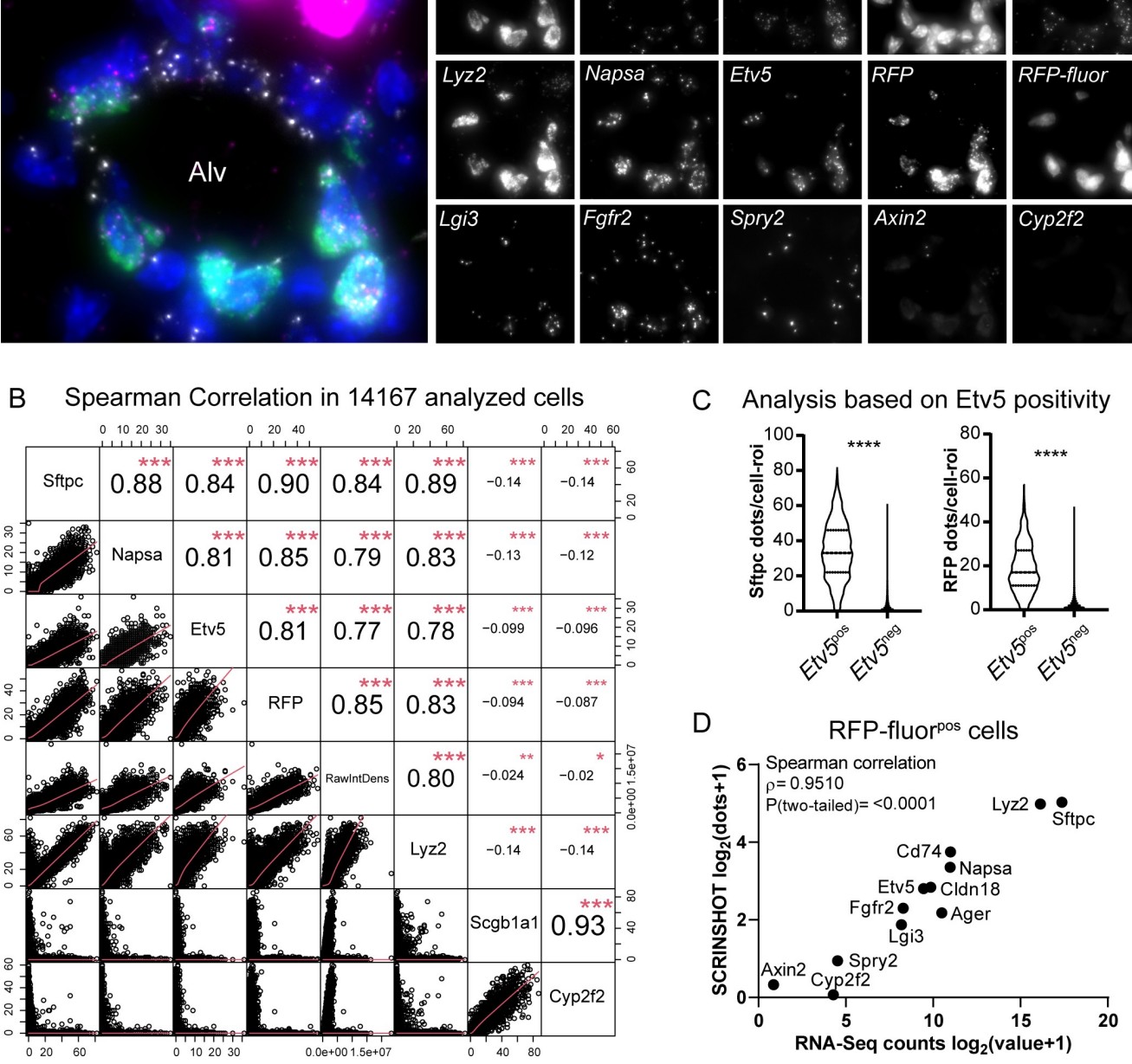

**Fig 5. *SCRINSHOT multiplex detection for AT2 selective genes indicates quantitative performance and high* correlation with the GSE118891 scRNA-Seq dataset [36].** (A) Representative image of an Alv region, showing the SCRINSHOT dots of 13 genes in addition to *RFP* transgene, RFP protein fluorescence, and the club cell marker *Cyp2f2* (negative control). Merge image includes *Sftpc* (green), *Cd74* (magenta), *Ager* (gray), and DAPI (blue). The rest are indicated with italics, and only single-channel images are shown. Scale bar: 10 μm. The data underlying this Figure section can be found in 10.5281/zenodo.3634561. (B) Spearman correlation analysis, in all analyzed cells (14,167) of the *Sftpc-CreER*^pos^;*Rosa-Ai14*^pos^ lung section, for the highly selective AT2 markers, *Sftpc*, *Napsa*, *Etv5*, and *Lyz2*. *RFP* SCRINSHOT-detected RNA transcripts and RFP protein fluorescence (raw integrated density) were also included to the correlation, as AT2 markers. Scgb1a1 and Cyp2f2 club cell markers were used as negative controls, anticorrelating with all other markers (negative coefficient values, ρ). (C) Violin plots of *Sftpc* (left) and *RFP* (right) SCRINSHOT dots, in *Etv5*^pos^ (>3 dots) and *Etv5*^neg^ (0–3 dots) alveolar cells. *Etv5*^pos^ $n = 1,431$ cells and *Etv5*^neg^ $n = 12,736$ cells. (D) Correlation plot of the indicated genes, between the average log₂(raw counts+1) values of 156 AT2 cells of GSE118891 and SCRINSHOT signal dots of 1,425 RFP fluorescence positive cells. The data underlying these Figure sections can be found in 10.5281/zenodo.3978632. Alv, alveolar; AT2, alveolar type 2; RawIntDens, raw integrated density; RFP, red fluorescent protein.

play a role in cystic fibrosis pathophysiology [52, 55]. They also provided the detailed transcriptomic state of additional cell types, like basal, tuft, and secretory cells, including club and 2 classes of goblet cells.

We tested the ability of SCRINSHOT to detect the aforementioned cell types and to analyze tracheal epithelial cell heterogeneity with spatial, single-cell resolution. We used a panel of selective markers for club, goblet, basal, tuft, and ionocytes, as identified in [52, 55]. The 29 analyzed genes included (1) *Scgb1a1*, *Scgb3a1*, *Il13ra1*, *Reg3g*, *Lgr6*, and *Bpifb1* as club-cell markers; (2) *Foxq1*, *Gp2*, *Pax9*, *Spdef*, *Tff2*, *Lipf*, *Dcpp3*, and *Dcpp1* as goblet cell markers; (3) *Trp5*, *Il25*, *Gng13*, *Six1*, *Alox5ap*, and *Sox9* as tuft cell markers; (4) *Foxi1*, *Tfcp2l1*, *Cftr*, and *Ascl3* as ionocyte markers; (5) *Ascl1* as an NE cell marker [51]; and (6) *Krt5*, *Pdpn*, and *Trp63* as basal cell markers. We also included *Muc5b* as a general secretory marker of the proximal airways [52, 55]. For assignment of cell positions, we utilized structural landmarks that separate the tracheal airway epithelium in 3 parts, the proximal, which extends until the end of the submucosal gland; the intermediate part, which spans 8 cartilage rings deeper; and the distal, which includes the remaining part of trachea epithelium, up to its bronchial bifurcation (carina). We also assigned positions to proximal intralobar airway epithelial cells, which extend up to the L.L3 branching point [56], and to distal airway epithelial cells, which are located at terminal bronchioles (Fig 6A).

For cell-type annotation, we used the combinatorial expression of top selective markers from published scRNA-Seq experiments to annotate the cells, as it is common in single-cell analyses [52]. We first selected the published top markers and assigned cells for marker positivity based on SCRINSHOT dots, above a certain threshold for each marker. The threshold was set at 10% of the maximum value of the marker gene dots, with the higher threshold being set to 3, which was applied for highly abundant genes with maxima over 31 signal dots. A cell was considered to belong to a category if it was positive for at least 2 of its selective markers and only expressed up to one marker of the other categories. For example, to consider a cell as basal, it should score positive for, at least 2 of the *Krt5*, *Pdpn*, and *Trp63*. Among the annotated basal cells, 82.9% had ≥1 dots for 2 of the markers and 17.1% for all 3. For ionocytes, we used *Ascl3*, *Foxi1*, *Cftr*, and *Tfcp2l1* as markers. The 42.9% of the annotated ionocytes showed ≥1 dots for 2 of the markers, and 57.1% were positive for 3 or more markers. For tuft cells, we used the *Gng13*, *Trpm5*, *Alox5ap*, *Six1*, *Sox9*, and *Il25* as markers. The 63.8% of the annotated tuft cells had ≥1 dots of 2 of the corresponding genes and 36.2% for 3 or more. For goblet cells, we used *Tff2*, *Gp2*, *Pax9*, *Lipf*, *Dcpp3*, *Dcpp1*, *Foxq1*, and *Spdef* as markers. The 41.9% of the annotated goblet cells had ≥1 dots of the *Tff2*, *Gp2*, *Pax9*, *Lipf*, *Dcpp3*, *Dcpp1*, *Foxq1* and *Spdef*, and 58.1% showed ≥1 dots for 3 or more of these genes. We considered a cell as club, if it was positive (>3 dots) for Scgb1a1 only. The 32.1% of the annotated club cells were positive for that marker, and they are mainly localized at the distal lung airway epithelium (Fig 6B and 6C). In the trachea and proximal airways, 12.8% of the club cells also had ≥1 dots of either *Scgb3a1*, or *Reg3g*, or *Bpifb1*, or *Il13ra1* or *Lrg6*, and 55.1% were positive for 2 or more markers (S8A Fig).

We sampled 1,068 cells in submucosal glands and 216 proximal trachea airway epithelial cells. In the intermediate part of the tracheal tube, we quantified 953 cells and in the distal portion, 1,164 cells. In the intralobar airway epithelium, we analyzed 551 cells in proximal and 484 in distal airways. SCRINSHOT detected all the previously described trachea cell types. Club cells comprised 7% of total cells in proximal trachea, and their proportion gradually increased to 77% toward the distal intralobar airways (Fig 6B). $Trp63^{pos}$, $Krt5^{pos}$, and $Pdpn^{pos}$ basal cells were primarily detected in the intermediate part of the trachea (21% of the counted cells) and became reduced towards the intralobar airways (Fig 6B, 6C and 6D). Tuft cells expressing *Trmp5*, *Gng13*, and *Alox* were exclusively found in the tracheal epithelium (Fig 6B,

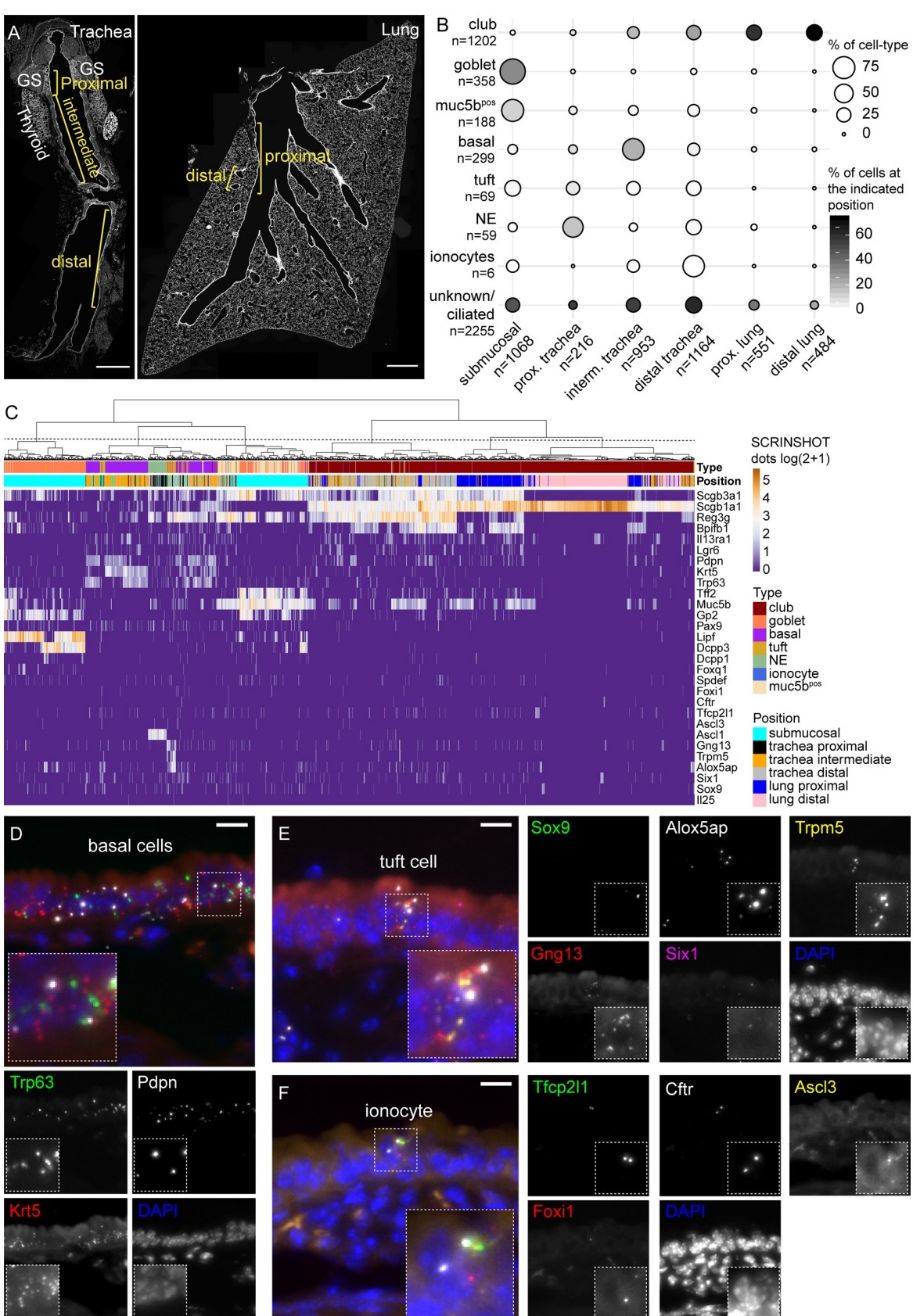

**Fig 6. Spatial mapping of tracheal cell types with SCRINSHOT.** (A) Overview of analyzed trachea and lung tissue sections using nuclear staining. Analyzed areas are indicated by brackets, corresponding to proximal trachea, GS, intermediate, and distal trachea, followed by proximal and distal lung airways. Trachea image includes also thyroid glands. Scale bar: 500 μm. (B) Balloon plot of the annotated cell types, at the analyzed positions. The balloon size indicates the percentage of the cell-type at the indicated position relatively to the total number of cells of the cell-type. The balloon intensity corresponds to the percentage of the specified cell-type, relative of the total number of cells at the indicated position. (C) Hierarchical clustering of the annotated cells. Heatmap shows the log$_2$(dots/cell + 1) gene values in the analyzed cells. (D) Characteristic example of basal cells in tracheal epithelium. *Trp63* (green), *Ktr5* (red), *Pdpn* (gray), and DAPI (blue). Scale bar: 10 μm. (E) Detection of a tuft cell in tracheal epithelium. *Sox9* (green), *Gng13* (red), *Alox5ap* (gray), *Trpm5* (yellow), *Six1* (magenta), and DAPI (blue). Scale bar: 10 μm. (F) An ionocyte in trachea airway epithelium, detected with *Tfcp2l1* (green), *Foxi1* (red), *Cftr* (gray), *Ascl3* (yellow), and DAPI (blue). Scale bar: 10 μm. The data underlying this Figure can be found in 10.5281/zenodo.3634561. GS, submucosal gland; RFP, red fluorescent protein.

6C and 6E). Ionocytes present a rare cell type (<1% of trachea epithelium) implicated in the pathogenesis of cystic fibrosis. We detected sparse ionocytes, expressing *Cftr* together with the transcription factors *Tfcp2l1*, *Ascl3*, and *Foxi1* [52, 55] in the tracheal airway epithelium (Fig 6B and 6F) and even more rarely in the submucosal glands (S10 Fig). Their restricted positioning in the tracheal and submucosal gland epithelium highlights the importance of these regions in cystic fibrosis caused by *Cftr* mutations in experimental models and in patients [57, 58].

The majority of the *Ascl1^pos^* NE-cells were detected in the proximal trachea, but as expected, we also detected positive cells, scattered along the airways (Fig 6B and 6C). Interestingly, 97% of the goblet cells were detected in the submucosal gland but not in the airway epithelium (Fig 6B and 6C and S11 Fig). In a hierarchical clustering of all annotated cells, the goblet cells were grouped together with the Muc5b^pos^ cells into 2 clusters (Fig 6C). In agreement with the identification of 65 goblet cells in the scRNA-Seq data of Montoro and colleagues [52], *Gp2* is detected in the majority of goblet cells. We also detected *Tff2^pos^ Muc5b^pos^* cells corresponding the described *goblet-1* sub-cluster and cells corresponding to the *Dcpp3^pos^ Lipf^pos^ goblet-2* sub-cluster.

Interestingly, SCRINSHOT revealed some additional spatial heterogeneity of *goblet-2* cells in the submucosal glands. We also noticed Gp2^pos^-positive cells, expressing high levels of either *Dcpp3* or *Lipf* (S11A Fig) and second, a small subset of the Dcpp3^pos^ cells also expressed *Dcpp1* (S11B Fig, arrowhead) [52].

To answer how many cells of a particular cell type also express markers for other cell types, we analyzed all the annotated cells and counted all signal dots for each of the analyzed genes. We then plotted the percentage of cells in each cell type expressing one or more dots of all genes (S8B Fig). Our analysis indicates that there is some co-expression of markers, which is more prominent for the club cell markers *Scgb1a*, *Scgb3a1*, *Reg3g*, *Bpfb1*, and *Muc5b*. This could be explained in 2 ways, either this marker combination and the assay are not specific or that this marker signature is selectively expressed at higher levels in club cells and at lower levels in the remaining cell types. To address this, we used the published dataset of Montoro and colleagues [52] and analyzed the cells with Seurat v3.1 [59] to retrieve and plot the percentages of positive cells per cell type, for all analyzed genes with SCRINSHOT (S8E Fig). In this analysis, we used the original annotation of the GSE103354 dataset. The heatmap of marker gene expression in different cell types, as indicated by the percentage of cell-type annotated cells expressing >0 Raw counts of the marker, was strikingly similar to the heatmap from SCRINSHOT, excluding the possibility that the detected co-expression of club genes in other cell types are a SCRINSHOT-associated artifact. The club cell markers *Scgb1a1*, *Scgb3a1*, and *Reg3g* are also expressed by all cell types, but their expression is higher in annotated club cells, with both methods. SCRINSHOT did not detect *Scgb1a1* in goblet cells, presumably due to the reduced padlock concentration to prevent signal saturation in the distal lung club cells. This explanation is consistent with the expected detection of the gene in the other cell types, that

express higher levels of the marker than goblet cells, in scRNA-Seq data (S8F Fig). To exclude the possibility that the low expression levels of the analyzed genes in droplet-based dataset are background, we examined the Fluorescence-activated cell sorting (FACS)-based dataset of the same paper, using *Cftr* as an example. Similarly, low levels of *Cftr* expression were detected in club, basal, and tuft cells, in addition to the higher expression levels of ionocytes (S8G Fig).

The majority of the introduced cell-type markers are selective but not unique for the corresponding cell types. The introduced annotation criteria are a strict but easy to apply, supervised approach to interrogate SCRINSHOT datasets and identify cells expressing combinations of cell-type markers. The omission of markers for cell types, that are anatomically detected in the analyzed areas is considered the major source of "not-annotated (NA)" cells. In the airway epithelium, the ciliated cells are considered the main source of NA cells. In the submucosal glands, we failed to annotate a larger number of cells because in this analysis, we excluded cells co-expressing club cell markers, like *Scgb3a1* and *Reg3g* (at lower levels than in club cells), together with goblet-cell-type markers (S8 Fig). In addition, submucosal glands contain other cell types, like ciliated, myoepithelial, and neural cells [60, 61] for which, we had no markers.

To provide a complementary, more holistic icon of the SCRINSHOT performance, we used hierarchical clustering, as an unsupervised method for grouping of the analyzed cells. For this type of analysis, we included all the cells that express more than 3 dots of any combination of the analyzed genes. Many of the previously excluded cells, as NA, were now clustered together with the annotated cell types, suggesting that the utilized annotation criteria including both positive and negative conditions are too strict but facilitate the easy interrogation of large datasets (S8C Fig). Cluster 1 contained the majority of club cells of proximal and distal intralobar lung airways and cluster 3, the club cells of the trachea. Cluster 2 and 4 included the submucosal gland goblet cells, in addition to many "NA" cells that expressed goblet cell markers, in addition to the club cell markers *Scgb3a1* and *Reg3g*. Cluster 5 was the most heterogeneous, because it contained the annotated ionocytes and basal and NE cells. Interestingly, this clustering placed ionocytes together (insert in S8C Fig), similarly to the *Ascl1*$^{pos}$ NE-cells. Finally, cluster 6 contains the majority of the NA cells, which are localized in intermediate and distal tracheal airway epithelium. Its left branch likely corresponds to ciliated cells, because of the absence of *Scgb1a1* expression. The presence of *Reg3g* and low *Scgb3a1* agrees with the scRNA-Seq data (S8D and S8F Fig). The right branch contains the rest of the tracheal club cells, which are negative for the *Bpifb1*, similarly to the scRNA-Seq data (S8D and S8F Fig).

The spatial analysis of epithelial cell types in the trachea and lung airways demonstrate the utility of SCRINSHOT in the localization of rare cell types in a complex tissue. Importantly, the choice of appropriate panels of cell-type markers is essential for detection of the cells of interest, as in any targeted method. The fact that no unique but only selective markers were identified and used for the spatial mapping of the cell types [52] reveals the high heterogeneity of respiratory epithelium and points out the importance of combinatorial detection of several markers to spatially track the cells in the tissue.

## Spatial mapping of airway and alveolar cells

We used the expression values of 15 genes in 14,167 cells to generate a spatial map of macrophages and AT1 and AT2 cells in the alveolar compartment and club and NE cells in the airways (S12 Fig). We annotated an airway cell as secretory if it was *Scgb1a1*$^{pos}$ *Cyp2f2*$^{pos}$ *Ascl1*$^{neg}$ and NE if it expressed *Ascl1*. In the alveolar compartment of the *Sftpc-CreER*$^{pos}$-*Rosa-Ai14*$^{pos}$ lung, we annotated 1,679 *Sftpc*$^{pos}$ cells as AT2. More than 99% of them also contained more than 3 dots of *Lyz2*, another AT2 marker [62], which is also expressed by lung inflammatory

cell types [39, 63]. The 91.8% of the AT2 cells were also scored positive for *Cd74*. This gene is significantly enriched in AT2 cells, but it is also expressed in hematopoietic lineages [64]. Additionally, SCRINSHOT analysis distinguished a group of 588 *Lyz2*$^{pos}$ *Cd74*$^{pos}$ *Sftpc*$^{neg}$, which are probably macrophages. We also detected 312 cells positive only for *Lyz2* and 450 expressing only *Cd74*. This suggests that the combinatorial expression of 3 genes defines 4 cell populations in the alveolar compartment, AT2 cells, *Lyz2*$^{pos}$ *Cd74*$^{pos}$ macrophages and single *Lyz2*$^{pos}$ or single *Cd74*$^{pos}$ immune cells. For identification of AT1 epithelial cells, we used the expression of *Ager* [65]. We scored 1,129 alveolar *Ager*$^{pos}$ *Sftpc*$^{neg}$ *Scgb1a1*$^{neg}$ cells as AT1. This analysis argues that SCRINSHOT can readily distinguish known epithelial cell types in the lung airways and alveoli. Additional markers for other characterized cell types, such as endothelial cells and fibroblasts, could be also included in future analyses to facilitate the creation of complete spatial maps of cell types in various organs and help to elucidate gene-expression and cell-type distribution patterns in healthy and diseased tissues.

## Conclusion

SCRINSHOT simplifies multiplex, in situ detection of RNA in various tissues. The assay is optimized for high sensitivity and specificity. We present a robust analysis protocol with minimal requirements for extensive instrumentation or computational skills, which makes it user-friendly and accessible in many fields of biology. The protocol is suitable for validation of scRNA-Seq results in small- or large-scale experiments or for the analysis of gene expression changes upon genetic or chemical manipulations. Importantly, the cost of SCRINSHOT for 30 genes (3 padlock probes/gene) is estimated between 350 and 650 SEK (32–59 Euros) for each reaction, depending on the size of the tissue sections, and it is mainly driven by the cost of the enzymes (S3 Table). Its low cost, compared with other in situ hybridization methods or antibody detection, and the straightforward analysis protocol makes it suitable for routine laboratory use. Although SCRINSHOT multiplexity is reduced compared with "in situ sequencing" by ligation [23] or other barcode-based approaches [4, 66], the number of detection fluorophores or hybridization detection cycles can be increased to detect 50 genes per slide.

We based cell segmentation on nuclear border drawing, because we did not have any indication of the real cell borders. This is far from perfect, but we expect that the detected RNA molecules of a given cell will be closer to its nucleus than to other neighboring nuclei. The 10-μm-thick sections allow partial overlapping of nuclei and cells, resulting in false registration of dots to neighboring cells, but thinner sections will impact the quality of histology. Our approach of manual segmentation is the most time-consuming part of the analysis, and we believe that automatic segmentation will provide a future solution. At present, our attempts with open access programs like Ilastik (https://www.ilastik.org/) [67] or Broad Institute Cell-Profiler (https://cellprofiler.org/) [68] only scored a 50% success rate in segmentation of lung sections presumably because of the elongated and overlapping cell shapes and the large cavities of the lung alveolar compartment. Recent advances in bioimage analysis indicate that artificial intelligence (AI) approaches can be utilized as an automated, reproducible methodology to solve the problem of cell and nuclear segmentation, but still there is no applicable user-friendly algorithm, as described in [69]. Interestingly, the majority of spatial transcriptomic methods use cultured cells and not tissue sections for validation, which allows easier identification of nuclear and cellular borders [4, 10, 70]. When tissue sections are analyzed, then the expression is measured in areas and not in single cells [23, 71]. Only a few methods, like seqFISH [31] and osmFISH, attempted a signal-assisted segmentation of the cells, but that requires the use of highly expressed genes for all cells and assumes that positive cells for the same marker will be clearly separated. We show that confocal microscopy and higher magnifications will improve

resolution, but cost will also increase. Cell membrane staining will be another complementary solution, as successfully used in [24], but the tissue treatment with organic solvents during hybridizations significantly compromises lipid- or antibody-mediated cell labeling.

The SCRINSHOT analysis of the mouse submucosal gland, trachea, and proximal-lung airway epithelium for 29 genes and distal lung for the expression of 15 genes demonstrates the capacity of the method to detect not only low abundance mRNAs but also rare cell types, like ionocytes and tuft and NE cells, facilitating the generation of cellular maps from complex tissues.

## Materials and methods

### Animals and histology

All experiments with wild-type (C57Bl/6J) mice were approved by the Northern Stockholm Animal Ethics Committee (Ethical Permit numbers N254/2014, N91/2016 and N92/2016).

For transgenic mouse experiments, *Sftpc-CreER*<sup>negative</sup>[45];*Rosa26-Ai14*<sup>positive</sup> [47] and *Sftpc-CreER*<sup>negative</sup>;*Rosa26-Ai14*<sup>positive</sup> individuals were used according to German regulation for animal welfare at the Justus Liebig University of Giessen (Ethical Permit number GI 20/10, Nr. G 21/2017). The recombination and *Sftpc*<sup>pos</sup> cell labeling of the *Sftpc-CreER*<sup>pos</sup>; *Rosa26-Ai14*<sup>pos</sup> lung was done by one subcutaneous injection of Tamoxifen on P1, and the tissues were collected on P21. An *Sftpc-CreER*<sup>neg</sup>;*Rosa26-Ai14*<sup>pos</sup> littermate treated as above and used as negative control, in addition to a wild-type P60 lung.

For lung tissue collection, the mice were anesthetized with a lethal dose of ketamine/xylazine. Lungs were perfused with ice-cold PBS 1X (pH 7.4) through the heart right ventricle to remove red blood cells. A mixture (2:1 v/v) of (PFA 4% (Merck, 104005) in PBS 1X pH7.4): OCT (Leica Surgipath, FSC22) was injected into the lung, from the trachea using an insulin syringe with 20-24G plastic catheter B Braun 4251130–01), until the tip of the accessory lobe got inflated, and the trachea was tied with surgical silk (Vömel, 14739). The lungs were removed and placed in PFA 4% in PBS 1X (pH 7.4) for 4 hours for P21 lungs and 8 hours for adult, at 4˚C in the dark, with gentle rotation or shaking. The other analyzed organs were simply placed in PFA 4% in PBS 1X (pH 7.4) for 8 hours.

The tissues were transferred to a new tube with a mixture (2:1 v/v) of (30% sucrose in PBS 1X [pH 7.4]):OCT at 4˚C for 12–16 hours, with gentle rotation or shaking. Thereafter, tissues were embedded in OCT (Leica Surgipath, FSC22), using specific molds (Leica Surgipath, 3803025), and frozen in a slurry of isopentane and dry ice. Tissue-OCT blocks were kept at −80˚C until sectioning. We cut 10-μm-thick sections with a cryostat (Leica CM3050S) and placed them on poly-lysine slides (Thermo, J2800AMNZ), kept at room temperature for 3 hours with silica gel (Merck, 101969) and then stored them at −80˚C for further use.

### Embryonic human lungs

Use of human fetal material from elective routine abortions was approved by the Swedish National Board of Health and Welfare, and the analysis using this material was approved by the Swedish Ethical Review Authority (2018/769-31). After the clinical staff acquired informed written consent by the patient, the tissue retrieved was transferred to the research prenatal material. The lung sample was retrieved from a fetus at 8.5 weeks post-conception. The lung tissue was fixed for 8 hours at 4˚C and processed as with mouse tissues.

### Probe design

A detailed description of the padlock probe design is provided in the S1 Text. Briefly, the PrimerQuest online tool (Integrated DNA Technologies: IDT) was used to select sequences of

Taqman probes (40–45 nucleotides) for the targeted mRNA. These sequences were then interrogated against targeted-organism genome and transcriptome, with Blastn tool (NLM) to guarantee their specificity. The Padlock Design Assistant.xlsm file was used to split the sequences in 2 and integrate them into the padlock backbone. Padlock probes were ordered from IDT as 5′-phosphorylated (to facilitate ligation) Ultramer DNA oligos, and their sequences are provided in S1 Table.

The 40–45 Taqman probe sequences were also used to prepare the fluorophore-labeled oligos. Using the IDT OligoAnalyzer tool, the length of the sequences was adjusted to Tm 56°C. To remove the fluorescent oligos after each detection cycle, we exchanged "T" nucleotides with "U" and treated them with Uracil-DNA Glycosylase (Thermo, EN0362). Detection oligos were labeled at their 3′-end with fluorophores and manufactured by Eurofins Genomics. The sequences of fluorophore-labeled oligos are provided in S2 Table.

## Pretreatments of the slides

Slides were transferred from −80°C to 45°C to reduce moisture. A post-fixation step with 4% PFA in PBS 1x (pH 7.4) was done, followed by washes with PBS-Tween20 0.05%. Permeabilization of tissues was done with 0.1 M HCl for 3 minutes, followed by 2 washes with PBS Tween-20 0.05% and dehydration with a series of ethanol. SecureSeal hybridization chambers (Grace Bio-Labs) were mounted on the slides, and sections were preconditioned for 30 minutes at room temperature (R/T) with hybridization-reaction mixture of 1X Ampligase Buffer (Lucigen, A1905B), 0.05 M KCl (Sigma-Aldrich, 60142), 20% deionized Formamide (Sigma-Aldrich, F9037), 0.2 µg/ul BSA (New England Biolabs, B9000S), 1 U/µl Ribolock (Thermo, EO0384), and 0.2 µg/µl tRNA (Ambion, AM7119). To block unspecific binding of DNA, we included 0.1 µM Oligo-dT30 VN.

## Padlock probe hybridization, ligation, and RCA

Hybridization of the padlock probes was done in the pretreatment hybridization-reaction mixture solution, omitting Oligo-dT30 VN and adding each padlock probe to 0.05 µM final concentration. We used 3 padlock probes for every targeted RNA species. For the highly abundant *Scgb1a1* mRNA, we used 0.01 µM of one padlock probe to minimize molecular and optical saturation. Hybridization included a denaturation step at 55°C for 15 minutes and an annealing step at 45°C for 2 hours. Not-hybridized padlock probes were removed by washes with 10% Formamide in 2X SSC (Sigma-Aldrich, S6639). To minimize the effect of the previously documented SplintR ligase nucleotide preferences [11, 30], padlock probe ligation was performed overnight (O/N) at 25°C using the SplintR ligase (NEB, M0375) at a final concentration of 0.5 Units/µl, T4 RNA ligase buffer (NEB, B0216), and 10 µM ATP, according to manufacturer recommendations (see New England Biolabs webpage).

RCA was done O/N at 30°C using 0.5 Units/µl Φ29 polymerase (Lucigen, 30221–2). The reaction mixture contained also 1X Φ29 buffer, 5% Glycerol, 0.25 mM dNTPs (Thermo, R0193), 0.2 µg/µl BSA, and 0.1 µM RCA primers (RCA Primer1: TAAATAGACGCAGTC AGT*A*A and RCA Primer2: CGCAAGATATACG*T*C). The "*" indicate Thiophosphate-modified bounds to inhibit the 3–5 exonuclease activity of Φ29 polymerase [22]. A fixation step with 4% PFA for 15 minutes was done to ensure stabilization of the RCA-products on the tissue. Sections were thoroughly washed with PBS Tween-20 0.05% before next step.

## Hybridization of detection oligos

The visualization of the RCA products was done with hybridization of the 3′-fluorophore-labeled detection oligos. The reaction mixture contained 2X SSC, 20% deionized Formamide,

0.2 μg/μl BSA, 0.5 ng/μl DAPI (Biolegend, 422801), and 0.4 μM FITC-labeled and 0.2 μM Cy3- and Cy5-labeled detection probes for 1 hour at R/T. Washes were performed with 10% Form-amide in 2X SSC, followed by 6X SSC. Tissues sections were dehydrated with a series of etha-nol, chambers were removed, and the slides were covered with SlowFade Gold Antifade Mountant (Thermo, S36936) and a coverslip.

After image acquisition, the coverslips were removed by placing the slides in 70% ethanol in 45˚C. Then, sections were dehydrated using a series of ethanol to mount the hybridization chambers. After tissue rehydration with PBS Tween-20 0.05%, the detection oligos were digested with Uracil-DNA Glycosylase for 1hour at 37˚C. The reaction mixture contained 1X UNG buffer, 0.2 μg/μl BSA and 0.02 Units/μl Uracil-DNA Glycosylase (Thermo, EN0362). Destabilized oligos were stripped off by thorough washes with 65% deionized Formamide at 30˚C. Multiple rounds of hybridization and imaging, as described in this section, were per-formed until all genes were imaged.

### Image acquisition

Images were captured with a Zeiss Axio Observer Z.2 fluorescent microscope (Carl Zeiss Microscopy GmbH) with a Colibri led light source, equipped with a Zeiss AxioCam 506 Mono digital camera and an automated stage, set to detect the same regions after every hybridization cycle. For the confocal analysis of NEBs, we used a Zeiss LSM780 confocal microscope, equipped with Plan-Apochromat 63X/1.40 oil lens. Images were captured with optimal set-tings, according to Zeiss Zen Black software default settings.

### Image analysis

The nuclear staining was used to align the images of the same areas between the hybridiza-tions, and multichannel *.czi files, containing the images of all genes, were created using Zen2.5 (Carl Zeiss Microscopy GmbH). The images were analyzed as 16-bit *.tiff files, without compression or scaling. Images were tiled using a custom script in Matlab (The MathWorks, Inc.). Manual nuclear segmentation was done with Fiji ROI Manager [72]. The nuclear ROIs were expanded for 2 μm with a custom CellProfiler script and considered as cells. The signal dots were counted in these cell ROIs using CellProfiler 3.15 [68], Fiji [73, 74] and R-RStudio [66, 75–78] custom scripts (https://github.com/AlexSount/SCRINSHOT).

### Thresholding

A cell-ROI size criterion was applied to remove the outliers with very small or big surface. In particular, only cells included between 2 standard deviations of the mean size of the analyzed cells were further processed. In SCRINSHOT, to consider a cell ROI positive for an analyzed gene, we used a threshold strategy. First, we determined the maximum numbers of signal dots per cell ROI for all analyzed genes. A cell ROI was considered as positive if it contained more than 10% of the maximum number of signal dots for the specific gene. The higher threshold was set to 3, which was applied for highly abundant genes with maxima over 31 signal dots.

### Curation of the data

In general, the 2-μm nuclear expansion provides an underestimation of real signal dots and provides satisfactory results for airway cells (e.g., Fig 3A' merge-images with cell-ROI out-lines), but the cellular segmentation of the alveolar region is more challenging mainly because of the irregular cell shapes and their overlap. This gave false-positive cell ROIs due to dots from adjacent true-positive cells being erroneously assigned to their neighbors. To reduce the

noise, signal dots of highly abundant RNA species, or the RFP protein fluorescence, of the *Sftpc-CreER*pos;*Rosa26-Ai14*pos lung, were used to visually inspect and remove the problematic cell ROIs from further analysis.

## Clustering

Annotated cells of submucosal gland, trachea, and lung airway epithelium were clustered using hclust package in R [79]. $Log_2$(dots/cell + 1) values were used to calculate Euclidean distances and clustering was done using ward.D2 method. Balloon plots were created by ggpubr package in R [80] and heatmaps with pheatmap package [81].

## Analysis of *Sftpc-CreER*pos;*Rosa26-Ai14*pos cells

For the identification of the RFPpos alveolar cell-ROIs in the *Sftpc-CreER*pos;*Rosa26-Ai14*pos lung, all analyzed alveolar cells from the RFPneg *Sftpc-CreER*neg;*Rosa26-Ai14*pos lung were used to determine the maximum RFP transgene fluorescence (in raw integrated density values) and set it as a threshold. The RFPpos and RFPneg cell-ROIs were curated, and the *Sftpc* signal dots were measured in them.

For the correlation of RFP protein fluorescence with SCRINSHOT *RFP* signal, the RFP protein fluorescence (in raw integrated density values) of all segmented cell ROIs from the analyzed tissue sections was correlated with the detected RFP signal dots in the same cell ROIs, using simple linear regression analysis in GraphPad Prism with default settings.

## Immunofluorescence

Tissue sections were prepared, using the same protocol as SCRINSHOT. We used 1% BSA in PBS 1X (pH 7.4) with 0.2% Triton X100 (blocking buffer) to block unspecific protein binding and we incubated with primary antibodies O/N at 4°C. Slides were washed with PBS 1X (pH 7.4) 3 times for 5 minutes, and we incubated them with secondary antibodies in blocking buffer, when necessary, for 1 hour at room temperature. After three 10-minute washes with PBS 1X (pH 7.4), we counterstained nuclei with 0.5 ng/µl DAPI (Biolegend, 422801) and mounted with ProLong Diamond Antifade Mountant (Thermo, P36961). We used the anti-proSftpc rabbit polyclonal antibody (Seven Hills, WRAB-9337, dilution 1:1,000), the Alexa Fluor 488 anti-CC10 (E-11) mouse monoclonal antibody (Santa Cruz Biotechnology, Inc., sc-365992, dilution 1:500), the Alexa Fluor 555 anti-E-Cadherin mouse monoclonal antibody (BD Biosciences, 560064, dilution 1:100), and the Alexa Fluor 647 anti-CGRP mouse monoclonal antibody (Santa Cruz Biotechnology, Inc., sc57053, dilution 1:250). For proSftpc visualization, we used the Alexa Fluor 488 donkey anti-rabbit secondary antibody (Thermo, A-21206, dilution 1:400).

## Comparison of SplintR and cDNA-based detection of RNA species

To compare the performance of padlock probe hybridization (1) directly on RNA (SCRINSHOT) and (2) after cDNA synthesis, we used sequential 10-µm-thick sections from adult mouse lungs, fixed for 8 hours (see "Animals and histology" section). cDNA-based approaches in earlier publications have only used fresh frozen tissues and pepsin or proteinase K tissue treatments to increase RNA accessibility [3, 23]. The padlock probes were designed to recognize exactly the same sequence of the analyzed genes. For the highly (*Scgb1a1*, *Sftpc*) and intermediate (*Actb*) expressed genes, we used only one padlock probe and for *Pecam1*, 3 probes because of its lower expression levels.

For cDNA synthesis, the RNA species of the tissue sections were transformed to cDNA by RT using random decamers. The RNA strands were degraded with RNaseH to let padlock probes hybridize to the corresponding cDNA sequences. Padlock probe ligation was done with Ampligase DNA ligase. All other steps were done according to the provided SCRINSHOT protocol.

For cDNA synthesis, we used 20 U/μl of the SuperScript II Reverse Transcriptase (Thermo, 18064014), 1X SuperScript II RT buffer (Thermo), 0.5 mM dNTPs (Thermo, R0193), 10mers-random primer (Thermo), 0.2 μg/μl BSA (NEB, B9000S), and 1 U/μl RiboLock RNase Inhibitor (Thermo, EO0384), at 42°C, O/N. The slides were post-fixed with 4% (w/v) PFA in PBS 1X (pH 7.4) at RT for 30 minutes, following by 6 washes with PBS-Tween 0.05%.

## Evaluation of SCRINSHOT specificity using mutated padlock probes

The experiment was done according to the default SCRINSHOT protocol, using 0.01 μM of *Scgb1a1* padlock probes and 0.05 μM of the *Actb*. The "mismatch" probe had the same sequence as the normal *Scgb1a1* padlock probe with a C>G substitution at the 5′- ligation site, and the "3'-scrambled" probe had the same 5′-arm, but the 3′-arm was scrambled. *Actb* padlock probe was used as internal control to calculate the *Scgb1a1/Actb* ratios. The *Scgb1a1/Actb* ratios of cell ROIs with zero *Actb* signal dots were considered as zeros. The detection of all *Scgb1a1* RCA-products was done using a detection oligo, which recognizes the padlock backbone of *Scgb1a1* but not of *Actb*. *Actb* RCA-product was detected by a detection oligo, which recognizes its gene-specific sequence (S2 Table). The *Scgb1a1/Actb* fluorescence ratios were calculated using the raw integrated densities of the 2 genes in each cell ROI. For additional evaluation of SplintR ligase fidelity, we introduced a C>T substitution at the 3′- ligation site of *Cyp2f2* padlock probe1 and a T>C substitution at the 5′- ligation site of *Etv5* padlock probe 5. *Actb* padlock probe was used as internal control. All padlock probes were used at 0.05 μM concentration in reaction mix. We did the experiment as for *Scgb1a1*, using sequential tissue sections.

## Estimation of SCRINSHOT dot size of *Etv5* and *Cyp2f2*

To experimentally measure the SCRINSHOT signal dots, we used the unsaturated genes Etv5 and Cyp2f2, which are expressed in the alveolar and airway epithelium, respectively. Using a custom CellProfiler [68] script ("colored_dots_images.cpproj"), we analyzed the SCRINSHOT signals in a *Sftpc-CreER*[neg];*Rosa26-Ai14*[pos] lung area and created images with identified dots, being colored according to the default CellProfiler colormap. Then, we measured the occupied surface of the colored dots, with "color threshold" in Fiji, using the "dot_size_measurement. ijm" custom script [74].

## Evaluation of SCRINSHOT specificity using an antisense competitor

To compete the binding of Scgb1a1 padlock probe on its target transcript, an antisense competitor (A), which recognizes the Scgb1a1 mRNA between nucleotides 316–378 and masks the padlock probe hybridization site, was used in the same hybridization mixture with the *Scgb1a1* padlock probe (P) at different ratios: (1) P:A = 1:0, (2) P:A = 1:1, and (3) P:A = 1:5, keeping the *Scgb1a1* padlock probe concentration to 0.05 μM.

## Correlation with SCRINSHOT with published single-cell RNA sequencing datasets

The GSE118891 dataset was used to retrieve gene expression values (raw counts) of all AT2 and club cells, according to the cell annotation of provided metadata file [36]. The genes of

interest were selected and $\log_2(\text{dots}+1)$ transformed, and their mean values were calculated. Similarly, the SCRINSHOT signal dots per cell-ROI were $\log_2(\text{dots}+1)$ transformed. Pearson correlation analysis was done using GraphPad Prism.

The GSE103354 droplet dataset was analyzed with Seurat v3.1 [59] package in R [77], with 2,000 most variable genes and the first 27 principle components. The original annotation of the dataset [52] was used for cell-type gene expression analysis. We retrieved the percentage of positive cells (raw counts $> 0$) for the analyzed genes, using MAST differential expression analysis [82]. TPM values of GSE103354 full-length dataset were used to examine which cell types have been detected with *Cftr* sequencing reads, based on the original annotation of the dataset.

## Statistical analysis

All statistical analyses were done with GraphPad Prism, using nonparametric tests, because the SCRINSHOT data do not follow canonical distributions. Multiple comparisons were done using ANOVA Kruskal–Wallis multiple comparison test, without multiple comparison correction ("*": $P \leq 0.05$, "**": $P \leq 0.01$, "***": $P \leq 0.001$, "****": $P \leq 0.0001$). For pairwise comparisons, the statistical analysis was done using Mann–Whitney nonparametric *t* test ("*": $P \leq 0.05$, "**": $P \leq 0.01$, "***": $P \leq 0.001$, "****": $P \leq 0.0001$). Spearman correlation, with GraphPad Prism, was used to examine the correlation between SCRINSHOT and scRNA-Seq data. Spearman correlation, with "PerformanceAnalytic" R-package [83], was used to examine the correlation between SCRINSHOT-analyzed genes in AT2 cells.

## Availability of data and materials

The datasets and analysis files of the current study have been deposited at Zenodo repository (DOI: 10.5281/zenodo.3634561 and 10.5281/zenodo.3978632). All scripts are available at https://github.com/AlexSount/SCRINSHOT.

## Supporting information

**S1 Fig. Comparison of the SplintR-based (SCRINSHOT) and the cDNA-based in situ hybridization assays for high, intermediate, and low abundant genes in sequential PFA-fixed lung sections.** (A) Images of SplintR-based (SCRINSHOT) and cDNA-based in situ hybridization assays. DAPI: blue, *Scgb1a1*: green, *Sftpc*: gray, *Actb*: red, and *Pecam1*: cyan. Pink outlines show the 2-μm expanded airway nuclear ROIs, which are considered as cells. The square brackets indicate the magnified areas on the right. The "n" corresponds to the number of counted cells in large images. Scale bar: 100 μm. (B) Bar-plots of the analyzed gene signals, in the indicated tissue compartments, for SCRINSHOT and cDNA-based approaches. The differences between the 2 conditions are significant ($P < 0.0001$) for all analyzed genes. (C) Histograms of the analyzed genes. The y-axes indicate the percentage of the cell ROIs and the x-axes, the binned signal dots in each cell. In SplintR-condition, 350 cells localized in arw and 1,624 in alv compartment. In cDNA-condition, there are 295 airway and 1,706 alveolar cells. Analysis was done using raw images, with the same acquisition conditions and thresholds. Only for visualization purposes, signal intensity of *Scgb1a1* and *Sftpc* in cDNA-condition was set 5-times higher than SplintR. The data underlying this figure can be found in 10.5281/zenodo.3634561. alv, alveolar; arw, airway; cDNA, complementary DNA; PFA, paraformaldehyde-fixed; ROI, region of interest.
(TIF)

**S2 Fig. Analysis of single nucleotide mismatches at the ligation site of padlock probes, targeting the moderately expressed *Etv5* and *Cyp2f2*.** (A) SCRINSHOT signal dots of the control *Cyp2f2* padlock probe. ('A) indicates the area of square bracket in "A". (B) SCRINSHOT signal dots of the mutated (C>T at the 3′-arm) *Cyp2f2* padlock probe. ('B) indicates the area of square bracket in "B". The same *Actb* padlock probe was used in both conditions, as internal control, showing no statistically significant difference. DAPI: blue, *Cyp2f2*: red, *Actb*: gray. Scalebar: 10 μm. (C) Bar-plot of the SCRINSHOT dots per cell in all airway cells of the analyzed tissue sections (A and B). (D) Bar-plot of the *Cyp2f2/Actb* dot ratio in the control and mismatch conditions. (E) SCRINSHOT signal dots of the control *Etv5* padlock probe. (F) SCRINSHOT signal dots of the mutated (T>C at the 5′-arm) *Etv5* padlock probes. The same *Actb* padlock probe was used in both conditions, showing no statistically significant difference. DAPI: blue, *Etv5*: green, *Actb*: gray. Scalebar: 10 μm. (G) Bar-plot of the SCRINSHOT dots per cell, in all alveolar cells (E and F). (H) Bar-plot of the *Etv5/Actb* dot ratio in the control and mismatch conditions. Arrows indicate *Etv5*[pos] cells. The "n" values indicate the number of analyzed cells in each tissue section. The data underlying this figure can be found in 10.5281/zenodo.3978632. ns, not significant.
(TIF)

**S3 Fig. Analysis of SCRINSHOT signal saturation.** (A) Raw images of proximal (top) and distal (bottom) airway epithelium, showing the *Scgb1a1* SCRINSHOT dots. The arrows indicate areas with signal saturation, which are present only in distal airways. DAPI: blue, *Scgb1a1*: gray. Scalebar: 20 μm. (B) Violin plots of the average cell surface, as indicated by the 2-μm expanded nuclei and SCRINSHOT dot surfaces of *Etv5* (alveolar) and *Cyp2f2* (airway). Representative images of the recognized SCRINSHOT dots by CellProfiler custom script, that labels the identified dots according to the CellProfiler default colormap. The data underlying this figure can be found in 10.5281/zenodo.3978632.
(TIF)

**S4 Fig. SCRINSHOT application on mouse kidney and heart.** (A) Application of SCRINSHOT in adult mouse heart section, containing a vessel (v). The endothelial cell markers *Pecam1* (magenta), *Cldn5* (gray), and *Cdh5* (yellow) were detected close to the lumen of the vessel, where this cell type is normally located. *Acta2* (green) was detected at the inner thick part of the vessel wall, being consistent with the expression of the marker by vascular smooth muscle cells. Scalebar: 10 μm. Epithelial markers *Sftpc* (cyan) and *Scgb1a1* (orange) were not detected. The data underlying this figure section can be found in 10.5281/zenodo.3978632. (B) Representative image from adult mouse kidney sections shows signal for *Actb* (gray), *Napsa* (magenta), and *Lyz2* (cyan) but not *Scgb1a1* (green) and *Sftpc* (red). Scalebar: 50 μm. DAPI (blue) was used for nuclear staining in both images. The data underlying this figure section can be found in 10.5281/zenodo.3634561.
(TIF)

**S5 Fig. SCRINSHOT application on fetal human lung.** On the left, overview of a w8.5 whole left lung tissue section, showing SCRINSHOT signal for *SOX2* (green), *SOX9* (red), and *ASCL1* (gray). The square brackets correspond to the images on the right. DAPI (blue) was used as nuclear staining. Scale bar: 500 μm. (A) Representative image of proximal epithelium, which is highly positive of *SOX2* and *ASCL1* but not *SOX9*. (B) Representative image of highly *SOX9* positive distal epithelium. The data underlying this figure can be found in 10.5281/zenodo.3634561.
(TIF)

**S6 Fig. Comparison of detected *Calca* RCA-products in first and eighth hybridizations.** (A, B) Images on the left show the *Ascl1*^pos^ (gray) neuroendocrine cells of an airway neuroepithelial body, in relation to *Scgb1a1*^pos^ (red) club cells. DAPI: blue, scale bar: 10μm. (C) Note that neuroepithelial bodies are tightly packed cellular structures, as indicated by DAPI nuclear staining. The images on the right show the *Calca* RCA-products, detected in the first (D) and the eighth (E) detection cycles. (D'-F') Magnified areas of the indicated positions (brackets) of images D–F, respectively. (F) Overlay of *Calca* identified signal dots in first (cyan) and eighth (yellow) detection cycles, using the same threshold in CellProfiler. The data underlying this Figure can be found in 10.5281/zenodo.3634561. RCA, rolling circle amplification.
(TIF)

**S7 Fig. Analysis of the NEB cellular complexity, with confocal microscopy.** Image of the same NEB in S6 Fig, showing SCRINSHOT signal dots of *Scgb1a1* and *Ascl1*. (Top) Maximal orthogonal projection of 20 z-stacks. (Bottom) Individual z-stacks. Cell outlines (magenta) were based on *Scgb1a1* SCRINSHOT signal. DAPI: blue, *Scgb1a1*: yellow, *Ascl1*: gray. Scalebar: 5 μm. (B) Immunofluorescence of a representative NEB from an adult mouse lung. Club cells were stained with an anti-Scgb1a1 antibody and NE cells with an anti-CGRP antibody. Epithelial cell membrane was stained with an anti-E-cadherin antibody. (Top) Maximal orthogonal projection of 13 z-stacks. (Bottom) Individual z-stacks. Cell outlines, based on E-cadherin signal, are shown with magenta. Both images were acquired with optimal confocal microscopy settings, using a 63× oil-immersion lens. DAPI: blue, Scgb1a1: green, E-cadherin: red, CGRP: gray. Scalebar: 5 μm. The data underlying this Figure can be found in 10.5281/zenodo.3978632. CGRP, calcitonin gene-related peptide; NE, neuroendocrine; NEB, neuroepithelial body.
(TIF)

**S8 Fig. Analysis of SCRINSHOT performance in tracheal epithelium and correlation with the droplet-based and Smartseq2 scRNA-Seq datasets of GSE103354 [52].** (A) Table showing the percentage of the cells of the indicated cell types, which express 1, 2, and 3 or more markers of the indicated cell type. Numbers in the parentheses show the number of used markers for each cell type. (B) Heatmap of the percentage of cells of the annotated cell types, that express one or more SCRINSHOT dots of the analyzed genes. (C) Hierarchical clustering of all analyzed cells with more than 3 dots of any gene and combination, without application of any cell size criteria ($n$ = 4,115 cells). The heatmap shows the $\log_2$(dots+1) SCRINSHOT-detected dots for the corresponding genes. Red insert shows 4 clustered ionocytes. (D) Analysis of the droplet-based scRNA-Seq dataset, using Seurat v3.1 [59], with 2,000 most variable genes and the first 27 principle components. The original annotation of the dataset, from [52], was used for the gene expression analyses. Umap-plots show the gene expression pattern of cell-type representative markers. (E) Heatmap plot of the percentage of positive cells in each cell-type as indicated by Seurat (Raw counts > 0). (F) Violin plots of the scRNA-Seq droplet-based dataset, showing the gene expression levels of the used cell-type markers at the SCRINSHOT experiment. The dots indicate single cells. (G) Table of the *Cftr* TPM counts of SmartSeq2 scRNA-Seq dataset [52], that show low, but not zero, gene expression in basal, club, and tuft cells, in parallel to the high expression in the annotated ionocytes. The data underlying this figure can be found in 10.5281/zenodo.3978632. scRNA-Seq, single-cell RNA sequencing; TPM, transcripts per million.
(TIF)

**S9 Fig. Comparison of SCRINSHOT and scRNA-Seq SmartSeq2 sensitivities, based on the moderately expressed *Etv5*.** (A) Histogram of SmartSeq2 *Etv5* $\log_2$(counts+1) values (x-axis), in the GSE118891 annotated AT2 (black) and club (gray) cells. y-axis indicates percentage of

cells. Club cells were used to set the positivity threshold for *Etv5*, because 95% of them are found below value 4 (outside the continuous red line). Distribution of *Etv5* counts in AT2 cells was used to estimate the low-expressing cells that are found below one standard deviation (1.9) from *Etv5* count average (9.4) (dotted red line). (B) Same type of analysis, as in "A", for *Etv5* SCRINSHOT dots in RFP fluorescence positive cells of the *Sftpc-CreER*[pos];*Rosa-Ai14*[pos] lung section, shows that 96.5% of RFP-fluor[pos] cells have ≥1 Etv5 dots, and 14.8% of them are detected with low number of dots (below 1 standard deviation from *Etv5* SCRINSHOT dot average). The data underlying this figure can be found in 10.5281/zenodo.3978632. RFP, red fluorescent protein; scRNA-Seq, single-cell RNA sequencing.
(TIF)

**S10 Fig. Ionocyte in the submucosal gland.** (Top) overview of the analyzed submucosal gland, showing SCRINSHOT signal for ionocyte markers, *Tfcp2l1* (green), *Cftr* (gray), *Foxi1* (red), *Ascl3* (yellow), and DAPI (blue). Scale bar: 500 μm. (Bottom) Magnified area of the indicated square in overview image, showing a detected ionocyte in submucosal gland. Scale bar: 10 μm. The data underlying this Figure can be found in 10.5281/zenodo.3634561.
(TIF)

**S11 Fig. SCRINSHOT mapping of submucosal glands reveals spatial heterogeneity in goblet cell population.** Overview of the analyzed submucosal gland for the expression of 6 goblet cell markers shows their expression in submucosal gland but not airway epithelium. *Muc5b* is detected along the airway epithelium of the proximal trachea, indicating that it is a general proximal epithelial cell marker. *Tff2*: green, *Muc5b*: red, *Gp2*: gray, *Dcpp3*: cyan, *Dcpp1*: magenta, *Lipf*: yellow, DAPI: blue, and cell ROIs: pink. Scale bar: 500 μm. (A) Insert showing the previously described *Muc5b*[pos] *Tff2*[pos] goblet subtype (arrow) and the *Lipf*[pos] *Dcpp3*[pos] (asterisk). Single *Lipf*[pos] (hash) and *Dcpp3*[pos] (arrowhead) are detected in the same region, being positive for the general goblet cell marker *Gp2*. (B) Insert showing regionally restricted expression of *Dcpp1* in a subset of *Dcpp3*[pos] cells. Insert scale bar: 10 μm. The data underlying this figure can be found in 10.5281/zenodo.3634561. ROI, region of interest.
(TIF)

**S12 Fig. SCRINSHOT generates cell-type digital annotation maps of large tissue areas.** (1) Overview image of SCRINSHOT fluorescence signal dots for *Scgb1a1* (green), *Sftpc* (magenta), and *Ascl1* (gray) of a large area from P21 *Sftpc-CreER*[pos];*Rosa-Ai14*[pos] lung section, after Tamoxifen induction on P1 (RFP was not shown at that image). The image contains 14,167 manual segmented nuclei, which were expanded for 2 μm and considered as cells. (2) Spatial map of annotated cell types according to the indicated criteria. (A) Same airway area as in Fig 6B, showing club (*Scgb1a1*: green and *Cyp2f2*: yellow) and NE-cell (*Calca*: magenta and *Ascl1*: gray) markers, in an airway position with a neuro-epithelial body (NEB). (A') Cell-type digital annotation of the area corresponding to "A". The "*" indicate *Ascl1*[pos] *Calca*[neg] cells Club cells: green and NE-cells: red. (B) Same alveolar area as in Fig 6A showing *Ager*[pos] (gray) *Sftpc*[neg] (green) *Cd74*[neg] (magenta) AT1 cells (arrow), *Ager*[low] *Sftpc*[pos] *Cd74*[low] AT2 cells (asterisks), and *Ager*[neg] *Sftpc*[neg] *Lyz2*[pos] *Cd74*[high] macrophages (mΦ, arrowhead). (B') Cell-type digital annotation of the area corresponding to "B". AT1 cells: gray, AT2 cells: magenta and macrophages: white. All not annotated cells are depicted with blue. Scale bar: 200 μm. The data underlying this figure can be found in 10.5281/zenodo.3634561.
(TIF)

**S1 Table. Summary of the utilized padlock probes.**
(PDF)

**S2 Table. Summary of the utilized fluorophore-labeled-oligos, which have been used for the detection of RCA products.**
(PDF)

**S3 Table. Detailed analysis of SCRINSHOT cost (sheets 1 and 2) and information about the used reagents and lab-ware.** RCA, rolling circle amplification.
(XLSX)

**S1 Text. Detailed protocol of all the steps of SCRINSHOT.**
(PDF)

## Acknowledgments

We would like to thank Dr. Qi Dai for the thorough reading and suggestions on the detailed SCRINSHOT protocol (S1 Text).

## Author Contributions

**Conceptualization:** Alexandros Sountoulidis, Werner Seeger, Mats Nilsson, Christos Samakovlis.

**Data curation:** Alexandros Sountoulidis, Andreas Liontos.

**Formal analysis:** Alexandros Sountoulidis, Andreas Liontos, Hong Phuong Nguyen.

**Funding acquisition:** Christos Samakovlis.

**Investigation:** Alexandros Sountoulidis, Hong Phuong Nguyen.

**Methodology:** Alexandros Sountoulidis, Andreas Liontos.

**Project administration:** Alexandros Sountoulidis, Christos Samakovlis.

**Resources:** Alexandra B. Firsova, Athanasios Fysikopoulos, Erik Sundström, Christos Samakovlis.

**Software:** Alexandros Sountoulidis, Andreas Liontos, Xiaoyan Qian.

**Supervision:** Alexandros Sountoulidis, Werner Seeger, Mats Nilsson, Christos Samakovlis.

**Validation:** Alexandros Sountoulidis, Andreas Liontos.

**Visualization:** Alexandros Sountoulidis.

**Writing – original draft:** Alexandros Sountoulidis, Hong Phuong Nguyen.

**Writing – review & editing:** Andreas Liontos, Hong Phuong Nguyen, Alexandra B. Firsova, Athanasios Fysikopoulos, Xiaoyan Qian, Werner Seeger, Erik Sundström, Mats Nilsson, Christos Samakovlis.

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
