## [Editor Report · Decision Letter 0]

31 Jan 2020

Dear Dr Sountoulidis, 

Thank you for submitting your manuscript entitled "SCRINSHOT, a spatial method for single-cell resolution mapping of cell states in tissue sections" for consideration as a Methods and Resources by PLOS Biology.

Your manuscript has now been evaluated by the PLOS Biology editorial staff, as well as by an academic editor with relevant expertise, and I'm writing to let you know that we would like to send your submission out for external peer review.

Please re-submit your manuscript within two working days, i.e. by Feb 04 2020 11:59PM.

Kind regards,

Roli Roberts

Senior Editor

PLOS Biology

---

## [Decision Letter · Decision Letter 1]

28 Feb 2020

Dear Dr Sountoulidis,

Thank you very much for submitting your manuscript "SCRINSHOT, a spatial method for single-cell resolution mapping of cell states in tissue sections" for consideration as a Methods and Resources at PLOS Biology. Your manuscript has been evaluated by the PLOS Biology editors, an Academic Editor with relevant expertise, and by three independent reviewers.

The reviews of your manuscript are appended below. You will see that the reviewers find the work potentially interesting. However, while reviewers #2 and #3 are broadly positive, reviewer #1 is highly critical of your benchmarking and of the apparent performance of SCRINSHOT. We discussed these rather divergent reviews with the Academic Editor, and they were persuaded by reviewer #1 that the readers must be more thoroughly convinced of the merits of your method, by improved benchmarking against a more relevant suite of approaches. In addition, if the claimed advantages include ease of implementation and low cost, then these need to be more explicitly laid out and/or quantified.

Based on reviewers' comments and following discussion with the Academic Editor, I regret that we cannot accept the current version of the manuscript for publication. We remain interested in your study and we would be willing to consider resubmission of a comprehensively revised version that thoroughly addresses all the reviewers' comments, and especially those of reviewer #1. We cannot make any decision about publication until we have seen the revised manuscript and your response to the reviewers' comments. Your revised manuscript would be sent for further evaluation by the reviewers.

We appreciate that these requests represent a great deal of extra work, and we are willing to relax our standard revision time to allow you six months to revise your manuscript.

We expect to receive your revised manuscript within 6 months. Please email us (plosbiology@plos.org) if you have any questions or concerns, or would like to request an extension. At this stage, your manuscript remains formally under active consideration at our journal; please notify us by email if you do not intend to submit a revision so that we may end consideration of the manuscript at PLOS Biology.

**IMPORTANT - SUBMITTING YOUR REVISION**

*Resubmission Checklist*

*Published Peer Review*

*PLOS Data Policy*

*Blot and Gel Data Policy*

Sincerely,

Roli Roberts

Senior Editor

PLOS Biology

REVIEWERS' COMMENTS:

Reviewer #1:

In their manuscript, Sountoulidis and colleagues report a technology called Scrinshot, a spatial method for mapping single cell states in tissue. The main merit of this in situ hybridization technology is low cost and multiplexing capacity. However, the sensitivity is not well characterized, the specificity appears poor, and the overall performance of generating single cell profiles is very low. The dataset of mapped cells does not appear close to the purported sensitivity of cell identification based on representative images provided of different cell types, and the profiling approach cannot even be applied to tissues or regions where cells are not of uniformly cuboidal morphology, like lung alveoli. Finally, there is considerable manual manipulation with processing of the primary fluorescent data with a puncta thresholding approach that is not validated, and adjudication of autofluorescence is required, introducing the possibility of bias.

Major criticisms

1) The sensitivity is benchmarked against surfactant C which is perhaps the most highly expressed gene in the lung. Despite the massive expression levels of this transcript, no signal was detected in 1.1% of the genetically marked alveolar type II cells. While sensitivity was not assessed for moderate or lowly expressed genes, the images provided show very few puncta (on the order of 2 or 3) for many transcripts. For instance, in Supplemental Figure 2B, there is a large heart blood vessel which is only sparsely decorated with puncta for PECAM1.

2) The authors provide some experiments showing that there is no signal when either the padlock cannot be ligated or the SplintR ligase is omitted. While these experiments demonstrate that a circle must be formed in order for signal to be produced, they do not substitute for control experiments showing that there is no significant off-target hybridization. The authors do perform a blocking oligonucleotide experiment for Secretoglobin 1a1 which has a nice result. However, when they actually quantify the specificity using surfactant C, this reveals a 7.4% false positive rate, which in my opinion is far too high to be acceptable for any staining technique. Supplemental Figure 4A-B encapsulate how poor the specificity of Scrinshot is, where most cells in a cluster that express the neuroendocrine marker Ascl1 (in panel A) co-express the Club cell marker Secretoglobin 1a1 (in panel B). Clearly, one of the markers is spuriously labeling the cells, since they should be complementary, rather than co-expressed. Another striking example is the type I marker AGER in Figure 6A for which puncta are shown in all epithelial cells, including most of the surfactant C positive cells which are putative type II cells. The authors seem to be aware of the extent of this specificity problem, because in their definitions of each cell type, they permit detection of one incompatible marker as long as there are two appropriate markers. This is highly contrived and betrays the high level of false positive signal produced by Scrinshot.

3) The heat map in Figure 7C includes a very large number of airway cells, but close examination of the profiles reveals very poor performance of Scrinshot, presumably due to the addition of poor segmentation on top of the low specificity. The populations are mostly distinguished by very high expression of a single canonical marker, and it does not appear the other transcripts influence the clustering, since looking at the expression of most markers across the cell types shows very poor correlation with the cell type in which it is known to be expressed. For instance, multiple club cell markers are inappropriately detected in basal and NE cells (e.g. Scgb1a1, 3a1, Reg3g). Also, apparently no ionocytes emerged (they are supposed to be coded blue in "Type" category) and CFTR expression is found scattered amongst cells belonging to most types.

4) The authors cite many of the multiplexed in situ and spatial mapping technologies that have been published over the last few years, but rather than comparing the performance of Scrinshot against any of these, they present data benchmarking Scrinshot against cDNA-based in situ, an early technology that was abandoned because of its poor performance. All contemporary approaches to multiplexed in situ hybridization rely on RNA detection, for instance branched-chain (RNAscope) and the two most closely related to Scrinshot (StarMAP and PLISH) that also employ rolling circle amplification. So, if the authors want to demonstrate that Scrinshot is an advance for the field, they must compare its performance to a current standard in the field, not an early-generation technology that has been far surpassed.

Reviewer #2:

The authors present a spatial-transcriptomic technique named SCRINSHOT that aimed to address current limitations of spatial-transcriptomics. Long (40 nt) gene-specific padlock probes are hybridized to PFA-fixed sections using a stringent protocol, followed by SplintR ligation of bound probes and phi-29 rolling circle amplification (RCA) to improve the signal-to-noise ratio. RCA products were detected using fluorescent-labelled oligos that bound the gene part of the padlock probe. Multiple rounds of hybridization allowed for the detection of ~ 30 genes (potentially 50 genes) in several thousand cells. The biological correlations used to validate SCRINSHOT are convincing and the level of detail in the protocol, from probe design to data analysis, is exemplary. Other techniques can detect more genes, some in 3D and at single molecule resolution, with examples of each referenced in the manuscripts introduction so this technique does not significantly outperform current state-of-the-art techniques in the field. However, the ease of SCRINSHOT implementation and lower equipment requirements will be of broad interest to labs that want to perform spatial-transcriptomics techniques and justifies publication in PlosBiology as a methods paper. However, I still have questions that need to be addressed before supporting acceptance of the manuscript.

Comments

● In my opinion the main novelty of this technique is ease of implementation. I suspect SCRINSHOT is cheaper compared to commercially available kits such as 10X Visium or RNAScope, providing a cost-effective way for labs to perform spatial transcriptomic. The authors echo this point in line 378; However, it is not sufficient to say SCRINSHOT is low cost without a detailed cost comparison vs commercially available spatial transcriptomics kits. The authors must include a detailed cost comparison so readers can decide if the reduced cost of SCRINSHOT justifies not using a commercial kit.

● The spatial mapping of AT2 and AT1 cells in the distal lung, shown in Supplementary Figure 7, differs from work published by Tushar Desai's lab showing AT1 cells adjacent to AT2 cells using IF and lineage tracing (DOI: 10.1126/science.aam6603). The simplest explanation is SCRINSHOT is not highly efficient resulting in false negatives. A detailed explanation about this discrepancy is required.

● No reference for Sftpc and Napsa as AT2 markers (e.g. https://www.ncbi.nlm.nih.gov/pmc/articles/PMC5135277/pdf/jciinsight-1-90558.pdf)

● Crowding of spots (proxy for a single RNA molecule) was described as an issue, yet in Figure 4 the number of dots was quantitated to perform correlation analysis with lineage-reporter expression with no explanation how crowding was overcome to allow such quantitation. Is crowding a gene specific issue (e.g. Scgb1a1)?

● Line 158: Similar Actb expression in all slides was mentioned in the text but not shown.

Reviewer #3:

In this manuscript, the authors report a novel approach to multiplexed in situ mRNA detection, based on a combination of hybridization using padlock probes, SplintR mediated ligation, RCA amplification and detection using fluorochrome-conjugated oligonucleotides. This approach allows detection of up to 30 probes with a good dynamic range for gene expression levels of the detected transcripts on PFA-fixed tissue sections of human and mouse lung. The paper is generally wel written, and the method is a major advance over existing methodologies such as MERfish and In Situ Sequencing, and will be of importance to generate spatial maps of cell types and states when combined with scRNA-Seq or snRNA-Seq data, which is an urgent need for atlassing studies of healthy or diseased tissues or in model organisms.

However, I also have some concerns and questions regarding some of the data, the analysis and the conclusions reached by the authors. In figure 2, the authors show data on the promiscuous ligation of the SplintR ligase, which might render aspecific signals. A single mismatch was introduced and shown to give strongly reduced signal, but not absence of signals. Given this effect, it would seem of relevance to explore this a bit further, and look at other mismatched probes (single mismatches in a slightly different sequence context, and involving other nucleotides), to allow a somewhat more systematic evaluation of the specificity of the technique.

In figure 4 and 5 the authors evaluate the quantitative detection power and the cell-type specificity of the SCRINSHOT methodology, using the RosaAi14 reporter mouse backcrossed to a Sftpc-CreER train to induce recombination of the reporter in type-2 cells after tamoxifen application. However, these two figures use semi-overlapping methodology or even data and are quite confusing as presented currently. In Figure 4, RFP is used to identidy AT2 cells, whereas in Figure 5 this is performed by Sctpc detection using SCRINSHOT. In figure 4, the correlation between SCRNSHOT RFP mRNA detection and RFP fluorescence (protein levels) is shown to be quite good, although in the CreER negative mice, some background in this channel was caused by red blood cell autofluorescence. It remains unclear, however, whether all cells were analyzed, or only the cells at a specific cut-off for the RFP fluorescence signal. Also, for clarity the authors should not refer to the RFP (transgene) fluorescence as 'endogenous'. 

In Figure 5, the authors go on to analyze AT2 cells by using Sftpc expression using SCRINSHOT detection as a means of AT2 cell identification (5A,B). This is initially confusing, so it could be better explained in the text on page 10. Just stating the relevant research question at the beginning of the paragraphs discussing the results presented in fig 4 and fig 5 would already be very helpful. In figure 5C, the authors use the same dataset but now define AT2 cells by RFP protein expression, and evaluate Sftpc expression by SCRINSHOT. Here, RFP+ Scftpc- cells are considered false negatives (for Sftpc expression) and RFP- Sftpc+ cells are considered false positives (for Sftpc expression). In Fig5D, his is reversed again (RFP false-positive and false-negative for Sftpc+ cells). This seems - as also mentioned by the authors - not the best way to identify AT2 cells, as lineage tracing using genetic markers has a considerable false-positive and false negative rate itself. Rather, using another AT2 marker as presented in fig 6 might be a good way to estimate false-positive and false-negative rates (see also comments at that figure). All in all ths figure and its data description is complex, confusing and in my opinion not the best analysis that could be done - some more evaluation of concordance between SCRINSHOT probes for AT2 cells might be helpful here, circumventing the use of the lineage trace. The conclusion of the authors should be about the eprformance of their technique, not of the lineage trace of the AT2 cells (which is now a significant confounding factor).

In figure 6, a number of AT2 probes are used, which are tested for correlation to a published AT2 single-cell dataset. Expresion was detected in the RFP+ cells. The selected scRNA-seq dataset seems arbritrary and small (a few hundred cells sequenced) - why was this specific dataset used? In addition to correlation to scRNA-seq data it would be very helpful to see correlation of AT2 markers to each other when detected by SCRINSHOT in the same tissue section. If serial hybridizations were done to generate the data for figure 6 (so it seems looking at the figures), his could be analyzed by the authors, and these data would be highly relevant to evaluate the performance of SCRINSHOT. 

In figure 7 the authors present a true spatial map using multiplexed SCRINSHOT. These data are very interesting and will likely represent a highly relevant technological advance to the field. However, some technical data are lacking that are relevant for the performance of the methods. For instance, what is the overlap between the markers for the same cell type? How many cells per cell type show 1 2 or 3 markers for that cell type? How many cells are stained by markers for different cell types? How were these dealt with in the analysis? How do the fractions of cells compare to available data from classical histology or singe-cell RNAseq studies? These sort of analyses would add greatly to the significance of the paper, especially since 2555 cells fal into the category of unknown/ciliated. What exactly does this mean? These cells are for instance very frequently observed in submucosal glands - what cell type would these be? The detection of specific cell types is a great sset in this technique, but if the majority of the cells is not assigned to a known cell type, he application of the technique will of course be limited. the authors need to discuss these cells, and the limitations of their approach. If the main problem is cell segmentation during the analysis, this should not be specific for this technique and authors would need to clarify this or compare to similar methods also hampered by cell segmentation accuracy during analysis. 

All in all, this is a highly interesting and relevant paper, and I feel a great asset for this section in Plos Biology of the authors can address the abovementioned concerns.

---

## [Decision Letter · Decision Letter 2]

18 Sep 2020

Dear Dr Sountoulidis,

Thank you for submitting your revised Methods and Resources entitled "SCRINSHOT, a spatial method for single-cell resolution mapping of cell states in tissue sections" for publication in PLOS Biology. I have now obtained advice from the original reviewers and have discussed their comments with the Academic Editor. 

We're delighted to let you know that we're now editorially satisfied with your manuscript. However before we can formally accept your paper and consider it "in press", we also need to ensure that your article conforms to our guidelines. A member of our team will be in touch shortly with a set of requests. As we can't proceed until these requirements are met, your swift response will help prevent delays to publication. Please also make sure to address the data and other policy-related requests noted at the end of this email.

IMPORTANT:

a) Many thanks for depositing the data underlying your Figures in Zenodo. Please could you clearly cite the Zenodo URL in the legends of all relevant main and supplementary Figures? e.g. "The data underlying this Figure can be found in 10.5281/zenodo.3634561"

b) Please could you make your title more declarative and without punctuation? Perhaps "SCRINSHOT enables spatial mapping of cell states in tissue sections at single-cell resolution." I should say that we're not in love with the name "SCRINSHOT," so if you can think of something more euphonious that would be great!

- a cover letter that should detail your responses to any editorial requests, if applicable

*Copyediting*

*Published Peer Review History*

*Early Version*

Sincerely,

Roli Roberts

Senior Editor,

rroberts@plos.org,

PLOS Biology

REVIEWERS' COMMENTS:

Reviewer #1:

After reviewing the revised manuscript and the authors' responses to my criticisms, I'm pleased to support publication in its current state.

The additional experiments they performed have gone a long way towards alleviating my concerns about the sensitivity and specificity of their technology. 

Reviewer #2:

We are happy with the authors' responses to our points now.

Reviewer #3:

The authors have gone to great lengths to address the issues raised in response to the first submission, and the manuscript has improved significantly as a result. All my major concerns have been accurately addressed. As a consequence of the review procedure and the careful revision, the inclusion of additional data and the careful discussion by the authors the manuscript now is quite long and offers a very detailed discussion. The manuscript is certainly acceptable in its current form, but might still benefit from a reread with the aim to improve legibility.

---

## [Editor Report · Decision Letter 3]

13 Oct 2020

Dear Dr Sountoulidis,

On behalf of my colleagues and the Academic Editor, Emma Rawlins, I am pleased to inform you that we will be delighted to publish your Methods and Resources in PLOS Biology. 

Early Version

PRESS 

Kind regards,

Alice Musson

Publishing Editor, 

PLOS Biology

on behalf of

Roland Roberts,

Senior Editor

PLOS Biology